# Screening, Identification, and Fermentation Condition Optimization of a High-Yield 3-Methylthiopropanol Yeast and Its Aroma-Producing Characteristics

**DOI:** 10.3390/foods13030418

**Published:** 2024-01-27

**Authors:** Yujiao Zhang, Qi Sun, Xiaoyan Liu, Rana Abdul Basit, Jinghao Ma, Zhilei Fu, Liujie Cheng, Guangsen Fan, Chao Teng

**Affiliations:** 1China Food Flavor and Nutrition Health Innovation Center, School of Food and Health, Beijing Technology and Business University, Beijing 100048, China; z18839793560@163.com (Y.Z.); s17321914603@163.com (Q.S.); liuxiaoyan@btbu.edu.cn (X.L.); basit98ft@gmail.com (R.A.B.); 15032049018@163.com (J.M.); chenglj17@163.com (L.C.); tc2076paper@163.com (C.T.); 2Department of Biology and Food Science, Hebei Normal University for Nationalities, Chengde 067000, China; 13833415159@163.com

**Keywords:** *Saccharomycopsis fibuligera*, Baijiu, tolerance, fermentation condition optimization, 3-methylthiopropanol, aroma characteristics

## Abstract

A high-yield 3-methylthiopropanol (3-Met) yeast Y1402 was obtained from sesame-flavored Daqu, and it was identified as *Saccharomycopsis fibuligera*. *S. fibuligera* Y1402 showed a broad range of growth temperatures and pH, as well as the maximum tolerance to glucose, NaCl, nicotine, and 3-Met at 50% (*w*/*w*), 15% (*w*/*v*), 1.2 g/L, and 18 g/L, respectively. After optimization using single-factor experiments, a Plackett–Burman design, a steepest ascent test, and a Box–Behnken design, the 3-Met yield reached 4.03 g/L by *S. fibuligera* Y1402 under the following optimal conditions: glucose concentration of 40 g/L, yeast extract concentration of 0.63 g/L, Tween 80 concentration of 2 g/L, L-methionine concentration of 5 g/L, liquid volume of 25 mL/250 mL, initial pH of 5.3, fermentation temperature of 32 °C, inoculum size of 0.8%, shaking speed of 210 rpm, and fermentation time of 54 h. The fermentation was scaled up to a 3 L fermenter under the optimized conditions, and the yield of 3-Met reached 0.71 g/L. Additionally, an aroma analysis revealed that the flavor substances produced by *S. fibuligera* Y1402 in sorghum hydrolysate medium was mainly composed of compounds with floral, sweet, creamy, roasted nut, and clove-like aromas. Therefore, *S. fibuligera* has great potential for application in the brewing of Baijiu and other fermented foods.

## 1. Introduction

3-Methylthiopropanol (3-Met) is a highly important sulfur-containing flavor compound that exhibits a low aroma threshold, presenting a pleasant meaty, grilled cheese, or sweet aroma at concentrations ranging between 1 and 3 mg/L [1,2,3]. It is a widely used edible flavor compound, and its market demand is growing rapidly [1]. Owing to its cheap cost, chemical synthesis is now the primary method for producing 3-Met; however, this synthesis has disadvantages as well, such as the use of hazardous raw materials, pollution during the synthesis process, and challenges in completely removing harmful byproducts [4]. These problems cannot meet the increasing consumer demand for environmentally friendly, green, healthy, and natural flavorings [4,5]. In this context, research efforts have shifted to the biotransformation synthesis, an ecologically safe, effective, and green approach for producing 3-Met [6]. Future predictions indicate that it will take precedence over other methods for the commercial synthesis of natural 3-Met.

At present, there are only a small number of natural microbial species that are involved in the synthesis of 3-Met. Only a few bacteria such as *Oenococcus oeni*, *Lactobacillus paracasei*, and *Lactobacillus curvatus*, and yeasts, such as *Saccharomyces cerevisiae*, *Kluyveromyces lactis*, and *Debaryomyces hansenii*, have been reported to produce 3-Met, but their yield was relatively low [1,4,7,8,9,10]. Except for *S. cerevisiae* SC408, *S. cerevisiae* SC57, *S. cerevisiae* S288C, *K. lactis* KL71, and the recently studied *Hyphopichia burtonii* YHM-G, the yield of 3-Met by other natural microorganisms were below 0.5 g/L [7,11,12,13,14]. To enhance the production of 3-Met by microorganisms, scientists have employed molecular biology techniques to examine the essential genes and enzymes responsible for 3-Met synthesis in native strains [2,15]. To maximize the output of 3-Met synthesis, these naturally existing strains were modified by overexpressing or shutting down genes, such as the aminotransferase genes *ARO8* and *ARO9* and the decarboxylase gene *ARO10* [14,15,16,17]. The greatest 3-Met output recorded in prior research was similarly attained by the genetically engineered strain, with a level of 4.38 g/L [18].

Even though genetically modified strains offer several potential benefits for the industrial production of 3-Met, consumers in some countries still do not consider them as natural products [7]. Consequently, there are some restrictions on how they may be used. For instance, 3-Met is a flavoring ingredient that plays a major role in the quality of many fermented foods, including cheese, wine, beer, soy sauce, Baijiu, and Huangjiu [8,19,20]. Research has shown that the 3-Met in these fermented foods mainly came from the microbial biotransformation, and its content has depended on the metabolic activity of microorganisms [19]. However, due to certain national policies, the use of genetically modified strains in the production and processing of fermented foods is strictly restricted [7]. Therefore, in order to improve the content of 3-Met in fermented foods, it is required by current rules to investigate and evaluate the fermentation properties of naturally occurring strains that produce large levels of 3-Met. Through controlling the metabolic processes of these indigenous microbes, it is feasible to elevate the 3-Met concentration and enhance the flavor profile of fermented foods [1,19]. Consequently, it is essential to identify and obtain more natural strains from various fermented food brewing conditions that provide elevated levels of 3-Met.

In this study, a high-yield 3-Met-producing strain was screened from the traditional Baijiu brewing process using traditional screening methods. The growth characteristics and fermentation conditions for 3-Met production were investigated. This study aims to increase the 3-Met content in Baijiu, hence improving its overall quality. In addition, this study will offer valuable strains and theoretical frameworks applicable to related fields.

## 2. Materials and Methods

### 2.1. Materials, Reagents, and Media

Sesame-flavored high-temperature Daqu (a starter for Baijiu production) at different production stages were provided by Guojing Group, Shandong, China Bandaojing Co., Ltd. (Zibo, China). 3-Met, L-methionine, and chromatography-grade methanol were purchased from Sigma-Aldrich (St. Louis, Mo, USA). Other chemical reagents were of analytical grade and can be commercially obtained unless otherwise stated. Luria–Bertani medium (LB medium for bacteria); yeast extract peptone dextrose medium (YPD medium for yeast) and potato dextrose agar medium (PDA medium for mold) for screening and culturing strains; Wallerstein laboratory nutrient agar medium (WL medium) for strain morphology identification; sorghum hydrolysate medium (SHM) for aroma compounds production; and original fermentation medium for 3-Met production were prepared as described previously [7,21,22]. 

### 2.2. Screening for Strain with High Yield of 3-Met

The isolation of strains was performed according to Fan et al. using the traditional plate coating method with LB, YPD, and PDA medium [23]. All isolated strains were individually cultured in their respective liquid culture media at 30 °C and 200 rpm. Bacteria and yeast were cultured for 36 h, while molds were cultured for 60 h. The cultured strains were adjusted to a cell or spore count of 1 × 10^6^ per milliliter using a hemocytometer. The process of producing 3-Met included transferring 0.1 mL of the corrected culture to a 250 mL shake flask with 50 mL of the original fermentation medium and incubating it for 48 h at 30 °C at 200 rpm, as described by Ma et al. [7]. Each strain was subjected to three parallel experiments. Strains showing the highest yield of 3-Met were selected for further research.

### 2.3. Identification and Biochemical Characteristics of S. fibuligera Y1402

#### 2.3.1. Morphological Observation 

Colony Morphology Observation: The yeasts were inoculated in YPD liquid culture medium and incubated for 24 h. The activated strains were streaked onto WL medium using the three-zone streaking method and incubated at 30 °C for 48 h in a constant humidity incubator. After incubation, colony morphology was observed. 

Cell Morphology Observation: A drop of sterile water was placed in the center of a sterilized microscope slide. Using an inoculating loop, a small amount of yeast cells from a single colony on the WL medium was picked and evenly spread in the sterile water on the slide. The cells were fixed by heating over an alcohol flame and were stained with methylene blue. A coverslip was placed on top, excess stain was removed using absorbent paper, and the slide was observed under an electron microscope for cell morphology observation.

#### 2.3.2. Physiological and Biochemical Identification

Physiological and biochemical tests were conducted by following the protocols outlined and described by Buchana et al. and Dong et al. [24,25]. These tests included carbon source fermentation, carbon source assimilation, nitrogen source assimilation, starch hydrolysis test, a methyl red test, an indole test, a urea test, a hydrogen sulfide test, a litmus milk test, a Voges–Proskauer test, a gelatin liquefaction test, and a citrate test.

#### 2.3.3. Molecular Biology Identification

Cultivation and Collection of Yeast Cells: The activated yeast cells were inoculated into YPD liquid medium and incubated at a temperature of 28 °C and a speed of 180 rpm for 48 h. The cells were then collected by centrifugation at 13,751× *g* for 10 min. 

Extraction of Yeast Genomic DNA and Amplification of 26S rDNA: The extraction of yeast genomic DNA followed the instructions of a fungal DNA extraction kit. The extracted yeast total DNA was used as a template for amplifying the 26S rDNA D1/D2 region using the universal primers NL1 (5′-GCATATCAATAAGCGGAGGAAAAG-3′) and NL4 (5′-GGTCCGTGTTTCAAGACGG-3′). The PCR reaction mixture consisted of 2.5 μL of LA PCR buffer, 1 μL of each primer, 2 μL of dNTP, 0.2 μL of LAtaq enzyme, 2 μL of DNA, and ddH_2_O to a final volume of 25 μL. The PCR amplification program was as follows: initial denaturation at 94 °C for 5 min, denaturation at 94 °C for 30 s, annealing at 58 °C for 30 s, and extension at 72 °C for 1 min for a total of 30 cycles, followed by a final extension at 72 °C for 10 min. The PCR products of yeast DNA were analyzed by electrophoresis on a 1% agarose gel. 

Sequencing and Homology Alignment: The purified 26S rDNA D1/D2 amplification products were sequenced to obtain the original sequences of the PCR fragments. The sequence alignment software BioEdit 7.0.9 (Borland Software Corporation, Scotts Valley, CA, USA) was used to align the sequences with the forward sequence. The corrected 26S rDNA D1/D2 region sequences were then compared with the corresponding sequences of known yeast species in the GenBank nucleotide sequence database using a BLAST search to determine the similarity.

#### 2.3.4. Determination of *S. fibuligera* Performance 

After being activated, *S. fibuligera* Y1402 was introduced into YPD medium with a 0.2% inoculum size to assess its growth temperature (ranging from 20 °C to 50 °C), pH levels (ranging from 1 to 14), sugar tolerance (33.3%, 37.5%, 41.2%, 44.4%, 47.4%, 50%, and 52.4%, *w*/*w*), NaCl tolerance (0%, 3%, 6%, 9%, 12%, 15%, 18%, and 21%, *w*/*v*), ethanol tolerance (0%, 3%, 6%, 9%, 12%, 15%, 18%, and 21%, *v*/*v*), nicotine tolerance (0.3 g/L, 0.4 g/L, 0.6 g/L, 0.7 g/L, 1.0 g/L, and 1.2 g/L), and 3-Met tolerance (ranging from 5% to 20%). The cultures were incubated in a shaking incubator at 28 °C and 180 rpm for 48 h. The OD_560_ was measured using a turbidity method. A blank control was performed using uninoculated medium under various cultivation conditions. 

#### 2.3.5. Production of Aromatic Compounds 

The activated *S. fibuligera* Y1402 were inoculated at a 0.2% inoculum size into the SHM. The cultures were incubated in a shaking incubator at a temperature of 28 °C and a shaking speed of 180 rpm for 48 h. The fermentation broth was then centrifuged at 4 °C and 13,751× *g* for 10 min. The supernatant was filtered through a 0.22 µm filter membrane and analyzed using headspace solid-phase microextraction–gas chromatography–mass spectrometry (HS-SPME–GC–MS) to detect the volatile aroma compounds produced by the yeast. A control group devoid of yeast cells was used. The previously disclosed method was used to treat the SHM medium.

### 2.4. Optimization of Fermentation Conditions for 3-Met Production

#### 2.4.1. Single-Factor Experiment

Based on the screening conditions mentioned above, a single-factor experiment was conducted to investigate the effects of medium composition (glucose concentration, yeast extract concentration, surfactant type and concentration, L-methionine concentration) and fermentation conditions (L-methionine addition time, temperature, initial pH, inoculum size, shaking speed, liquid volume, fermentation time) on the production of 3-Met by *S. fibuligera* Y1402 (Appendix A).

#### 2.4.2. Plackett–Burman Design

Nine factors (glucose concentration (A), yeast extract concentration (B), L-methionine concentration (C), temperature (D), initial pH (E), fermentation time (F), inoculum size (G), liquid volume (H), and shaking speed (I)) were chosen for optimization using the Plackett–Burman experimental design based on the significant difference analysis of the results of the single-factor experiment and experience. Each factor was tested at two levels, high (+1) and low (−1), and the experimental design was conducted using Design-Expert 11 software (Table 1).

#### 2.4.3. Steepest Ascent Test

Seven factors (initial pH, fermentation time, yeast extract concentration, inoculum size, L-methionine concentration, shaking speed, and temperature) that had a significant impact on 3-Met production by *S. fibuligera* Y1402, as determined from the Plackett–Burman experiment, were selected for optimization using the steepest ascent test. The direction of change for each factor was based on its correlation with 3-Met production by *S. fibuligera* Y1402 in the Plackett–Burman design. The gradient for each factor was determined based on the principles of the steepest ascent test, single-factor experiment results, and experimental experience. A total of five experimental gradients were set up (Table 2).

#### 2.4.4. Response Surface Analysis

Three components (fermentation time (A), yeast extract concentration (B), and initial pH (C)) that had the greatest influence were chosen for optimization utilizing the response surface analysis based on the significance order of factors impacting the yield of 3-Met from the Plackett–Burman design. The center point of the response surface analysis experiment was the highest point from the steepest ascent test. According to the principles of the Box–Behnken central composite design, a total of fifteen experiments with three factors and three levels, including three center point combinations, were designed using Design-Expert 11 software (Table 3).

### 2.5. Scale-Up Fermentation in Fermenter

The optimized medium composition and fermentation conditions were used for the scale-up fermentation experiment of *S. fibuligera* Y1402 in a 3 L fermenter. Considering the potential loss of 3-Met through exhaust emissions and the rapid growth of the yeast under high oxygen levels, the aeration rate of the fermenter was set at 0.5 air volume per culture volume per minute (vvm), the agitation speed was 300 rpm, the inoculum size was 0.8%, and the liquid volume was 1.8 L, taking cost into consideration. Other conditions were set according to the optimized conditions from the shake flask fermentation. Samples were taken every 8 h to measure the pH, cell density, and 3-Met concentration in the fermentation broth.

### 2.6. Analytical Methods

Yeast biomass was determined using the turbidity measurement (absorbance at 560 nm). The pH in the fermentation broth was measured using a pH meter. The analysis of volatile flavor compounds in the fermentation broth was performed using headspace solid-phase microextraction–gas chromatography–mass spectrometry, following the method described by Fan et al. [26]. The 3-Met concentration in the fermentation broth was determined using high-performance liquid chromatography, following the method described by Ma et al. [7].

### 2.7. Statistical Analysis

A one-way ANOVA (*p* < 0.05) with Tukey’s test was used to examine statistical differences in the evaluated techniques. Each experiment was performed in triplicate, and the experimental data were processed and plotted using Excel 2019 (Microsoft, Redmond, WA, USA), SPSS 24.0 (IBM Corp., New York, NY, USA), OriginPro 9.1 (OriginLab, Northampton, MA, USA), and Design-Expert 11 (Stat-Ease, Inc., Minneapolis, MN, USA).

## 3. Results and Discussion

### 3.1. Screening of High-Yield 3-Met Strains

A total of 94 strains were screened from high-temperature sesame-flavored Daqu using three different culture mediums, including 61 bacteria, 18 yeasts, and 15 molds. These strains were activated and cultured in the initial fermentation medium, and an analysis revealed that 35 strains were capable of producing 3-Met. This might be attributed to the specific sample of Daqu used, as previous studies have shown that 3-Met was an important flavor compound in sesame-flavored Baijiu [25]. Consequently, it is very possible that Daqu, which is utilized to promote fermentation and saccharification during the brewing of this particular variety of Baijiu, include 3-Met-producing strains. Among the tested strains (Table 4), most yeast strains were able to convert L-methionine into 3-Met, while most bacteria or molds were unable to do so. Among the strains capable of producing 3-Met, yeast generally exhibited higher concentrations compared with bacteria or molds, which was consistent with previous research [4,7]. This might be due to a higher enzymatic activity or expression levels of the enzymes involved in the Ehrlich pathway in yeast, which was responsible for the metabolism of L-methionine into 3-Met. Excitingly, 10 strains were found to produce 3-Met with a content exceeding 0.5 g/L, and all of these strains were yeast strains, which may also be related to the Daqu samples. Among them, yeast Y1402 produced the highest content of 3-Met, exceeding 2.0 g/L. Naturally, most strains that were able to produce 3-Met also resembled those from earlier studies in which the majority of microorganisms produced very little 3-Met [4,7]. To compare and analyze Y1402 with the previously isolated high-yield 3-Met yeast YHM-G within our team, we cultured these two yeasts under the same conditions in the initial fermentation medium. The results showed that both strains exhibited high 3-Met production capabilities, surpassing most of the reported high-yield 3-Met strains [7]. Furthermore, yeast Y1402 showed significantly higher 3-Met production compared with yeast YHM-G (Appendix A). Based on these results, we selected yeast Y1402 for further research. 

### 3.2. Identification of Yeast Y1402

The colony morphology of strain Y1402 on WL medium was shown in Figure 1a. The colonies were white and small and round with a raised center. The color of the medium changed from blue-green to yellow in the area where the strain grew. The surface of the colonies was smooth, moist, and easy to pick. The cellular morphology of strain Y1402 was shown in Figure 1b, with most cells being oval-shaped and a few appearing elongated with budding at the top (indicated in the box in Figure 1b). These observations indicated that strain Y1402 exhibited the typical colony morphology and microscopic structure of yeast. 

To determine the species of yeast Y1402, its physiological and biochemical characteristics were identified (Appendix A). The results of the sugar fermentation test showed that yeast Y1402 utilized glucose to produce acid and gas. It also produced acid but not gas when utilizing D-galactose, D-maltose, and sucrose. However, it could not utilize L-rhamnose, D-xylose, D-arabinose, and lactose to produce acid or gas. In the carbon source assimilation test, yeast Y1402 used D-maltose, D-galactose, glucose, sucrose, ethanol, and D-ribose as the sole carbon sources for growth, and the growth was good. Although it could use inulin and glycerol as carbon sources for growth, the growth was slightly poorer compared with ethanol and D-ribose. It could not utilize D-trehalose, D-raffinose, mannitol, L-rhamnose, D-xylose, D-arabinose, lactose, and D-sorbitol as the sole carbon sources for growth. The nitrogen source assimilation test results showed that yeast Y1402 used urea, ammonium sulfate, L-lysine, and L-phenylalanine as the sole nitrogen sources for growth, and the growth was good. It could also use sodium nitrite and potassium nitrate as the sole nitrogen sources for growth, but the growth was poor. The indole test, methyl red test, citrate test, and starch hydrolysis test were positive, indicating that yeast Y1402 can secrete tryptophanase, utilize tryptophan to produce indole, ferment glucose to produce pyruvic acid or succinic acid, grow using sodium citrate as the sole carbon source, and produce amylase to hydrolyze starch. However, it showed negative results in the hydrogen sulfide test, Voges–Proskauer test, urea test, and gelatin liquefaction test. Based on these physiological characteristics being similar to *S. fibuligera* M33, it was preliminarily identified as *S. fibuligera* [27].

The obtained 26S rDNA D1/D2 sequence of yeast Y1402 was subjected to a BLAST comparison on NCBI (https://www.ncbi.nlm.nih.gov/, accessed on 9 December 2023, accession number OR660109.1), and the results showed 100% similarity with *S. fibuligera* CBS:214.37 (accession number: MH867402.1), *S. fibuligera* LY41 (Accession number: KY705007.1), *S. fibuligera* KJJ81 (accession number: CP012816.1), and *S. fibuligera* ATCC36309 (accession number: CP015978.1). Combining morphological observations, physiological and biochemical characteristics, and molecular biology, yeast Y1402 was identified as *S. fibuligera*. *S. fibuligera* is common non-brewing yeast that plays an important role in fermented foods. It can produce enzymes such as lipase, protease, and amylase to promote the hydrolysis of nutrients in raw materials and produce flavor compounds such as ethyl acetate, phenethyl alcohol, isoamyl acetate, and phenethyl acetate [28,29,30,31]. It is important flavor-producing yeast in the fermentation process of food production.

### 3.3. Determination of S. fibuligera Y1402 Performance

The growth temperature, pH, and tolerance of *S. fibuligera* Y1402 to sugar, NaCl, ethanol, nicotine, and 3-Met is shown in Figure 2. The optimal growth temperature for *S. fibuligera* Y1402 was 25 °C, with a maximum tolerance of 40 °C (Figure 2a), which was consistent with most yeast and indicated its suitability for fermentation in products like Baijiu [32]. *S. fibuligera* Y1402 exhibited optimal growth at a pH of 4 and tolerated a wide range of pH values from 2 to 13 (Figure 2a), making it adaptable to various fermentation environments, such as the production of Baijiu and Huangjiu. Its lowest pH tolerance was similar to other yeast strains known for acid tolerance [33,34]. *S. fibuligera* Y1402 also demonstrated high tolerance to sugar, being able to grow even in the presence of 50% glucose concentration (Figure 2b), surpassing most yeast strains [7,26,35]. Regarding NaCl concentration, the yeast density of *S. fibuligera* Y1402 initially increased and then decreased as the NaCl concentration raised (Figure 2b). At a NaCl concentration of 3%, *S. fibuligera* Y1402 growth was promoted, indicating that its growth was favored under certain suitable NaCl conditions. However, when the concentration exceeded 3%, the high osmotic pressure inhibited the growth of *S. fibuligera* Y1402. Its maximum NaCl tolerance was as high as 15%, which was consistent with the NaCl tolerance of *Clavispora lusitaniae* YX3307 and higher than that of *Pichia kudriavzevii* YF1702, but lower than that of *H. burtonii* YHM-G [7,26,35]. In terms of ethanol tolerance, *S. fibuligera* Y1402 showed a decrease in cell density as ethanol concentration increased, and it hardly grew when the ethanol concentration exceeded 6% (Figure 2b). This indicated that *S. fibuligera* Y1402 was sensitive to ethanol, similar to *C. lusitaniae* YX3307 but lower than *H. burtonii* YHM-G [7,26]. Fortunately, the ethanol concentration in the production process of Baijiu was generally between 2 and 4%, suggesting that this yeast could be applied in Baijiu production [7,26,36]. To explore the potential application of *S. fibuligera* Y1402 in tobacco products, its nicotine tolerance was analyzed. The results showed that as the nicotine concentration increased, the inhibition of *S. fibuligera* Y1402 growth intensified. The maximum nicotine tolerance of *S. fibuligera* Y1402 was 1.2 g/L (Figure 2c), which was lower than the nicotine content in tobacco leaves (usually 1.5–3.5%) [37]. However, our research had revealed that *S. fibuligera* Y1402 could grow on tobacco leaves, possibly due to the low local nicotine content on the leaves and the influence of the structure of the tobacco leaves, which reduced the damage to yeast cells. To investigate its ability to produce 3-Met, the tolerance of *S. fibuligera* Y1402 to 3-Met was analyzed. As observed in Figure 2d, the growth of *S. fibuligera* Y1402 was increasingly inhibited with higher concentrations of 3-Met. The maximum tolerance to 3-Met was found to be 18 g/L, which was much lower than that of *H. burtonii* YHM-G and *S. cerevisiae* CEN.PK113-7D [5,7]. However, subsequent experimental results revealed that high tolerance to 3-Met only indicated the potential for the high yield of 3-Met but did not necessarily mean that higher tolerance would lead to a higher yield of 3-Met. Considering the above characteristics, *S. fibuligera* Y1402 has the potential for application in fermentation products such as Baijiu and tobacco.

### 3.4. Optimization of S. fibuligera Y1402 Fermentation Conditions for 3-Met Production

#### 3.4.1. Single-Factor Experiment 

Although the synthesis of 3-Met occurs through the Ehrlich pathway, glucose plays a significant role in providing energy for microbial growth and metabolic activities. It serves as a carbon skeleton that is an essential substance in microbial activities [26,38]. Therefore, the concentration of glucose has a certain influence on the synthesis of 3-Met by yeast. Increasing the glucose concentration at lower glucose concentrations facilitated the development and metabolism of *S. fibuligera* Y1402, which in turn facilitated the synthesis of 3-Met, as Appendix A illustrates. But when the concentration of glucose raised, *S. fibuligera* Y1402 grew quickly, which resulted in insufficient oxygen being supplied to the fermentation system. This raised the production of other higher alcohols that were excessive and might be hazardous to *S. fibuligera* Y1402, which in turn caused a reduction in the concentration of 3-Met generated. Additionally, it altered the metabolic route of *S. fibuligera* Y1402 in converting L-methionine to 3-Met [7,11,39]. From the experimental results (Appendix A), the maximum concentration of 3-Met was recorded in *S. fibuligera* Y1402 when the glucose content was between 40 and 55 g/L. This was more than the ideal glucose concentration for *S. cerevisiae* SC408 to produce 3-Met, but it was comparable with the glucose concentration needed for *H. burtonii* YHM-G to produce 3-Met [7,11]. This difference might be attributed not only to differences in the initial fermentation medium but also to the different yeast species. 

The Ehrlich pathway is an important pathway for the microbial synthesis of higher alcohols and other compounds. However, the nitrogen supply in the fermentation medium may readily affect this process. Specifically speaking on the synthesis of 3-Met in the present investigation, the Ehrlich metabolism route in yeast would be somewhat inhibited by the presence of nitrogen sources other than L-methionine, leading to a reduction in 3-Met synthesis [1,38]. However, research has shown that adding a certain quantity of yeast extract in the fermentation medium may encourage yeast to produce 3-Met [7]. This might be because yeast extract not only serves as an organic nitrogen source for microbial growth but also provides important growth factors such as amino acids, vitamins, and biotin for microbial growth and reproduction [3,7]. When yeast extract was present, the amount of 3-Met synthesized by *S. fibuligera* Y1402 was higher compared with when yeast extract was absent, as shown in Figure 3a. Moreover, a concentration of 0.4 g/L of yeast extract produced the maximum quantity of 3-Met. However, the quantity of 3-Met produced by *S. fibuligera* Y1402 decreased when the yeast extract concentration was raised over this threshold. This finding initially showed that *S. fibuligera* Y1402 benefited from the presence of yeast extract in the synthesis of 3-Met. Additionally, it showed that yeast extract functioned as both an Ehrlich pathway inhibitor and a growth promoter for yeast. When yeast extract was present in greater quantities, it blocked the Ehrlich pathway and functioned as a nitrogen source; however, at lower concentrations, it promoted yeast growth exclusively as a growth factor. The yeast extract concentration required to achieve optimum 3-Met synthesis by *S. fibuligera* Y1402 was comparable with that of *K. lactis* KL71 but lower than that of *H. burtonii* YHM-G, *S. cerevisiae* EC-1118, and *S. cerevisiae* SC408 [7,11,13,40]. This difference is mainly attributed to the metabolic variations among different yeast species.

Surfactants are a type of amphiphilic chemicals that can interact with microbial cell membranes to change their permeability through bilayer interactions. This influences the production of intracellular chemicals by microbes as well as their intake of nutrients. Furthermore, surfactants have the ability to control the amount of microbial enzymes secreted and affect the activity of these enzymes [41,42]. As a result, surfactants significantly affect the way that microbial cells use energy. Appendix A demonstrates that the other surfactant types had no discernible influence on the synthesis of 3-Met by *S. fibuligera* Y1402 with the exception of Tween 80 and Tween 60, which had a small boosting effect. The effect of surfactants on the formation of 3-Met by *H. burtonii* YHM-G, on the other hand, was different and might have resulted from small differences in the yeast cell’s membrane composition or the activity of associated Ehrlich pathway enzymes [8]. Moreover, Tween 80 did not significantly alter its action at different doses, even though it promoted the synthesis of 3-Met by *S. fibuligera* Y1402 in the range of 2–6.4 g/L (as shown in Appendix A). This indicated that Tween 80 had good fusion properties with the *S. fibuligera* Y1402 cell membrane, even at higher concentrations, without causing membrane disruption, electrolyte leakage, or metabolic imbalance, as described in other studies [43]. This was similar to the effect of Tween 80 on the synthesis of 3-Met by *H. burtonii* YHM-G [7]. Considering cost, a concentration of 2 g/L of Tween 80 was chosen, and no further optimization of its concentration was performed in subsequent experiments.

Through the Ehrlich pathway, 3-Met might be produced from L-methionine [8,44]. In summary, by undergoing transamination, decarboxylation, and reduction processes, 3-Met was produced from L-methionine [5]. Due to the fact that yeast synthesizes 3-Met through the Ehrlich pathway, the importance of L-methionine concentration in yeast for the synthesis of 3-Met is self-evident [13]. *S. fibuligera* Y1402 was shown to be unable to synthesize 3-Met in the absence of L-methionine, which was in accordance with other reports, as seen in Figure 3b [1,7]. Yeast was known to manufacture 3-Met via the Ehrlich pathway and was unable to use carbohydrates for de novo synthesis. The results also showed that the synthesis of 3-Met by *S. fibuligera* Y1402 was significantly impacted by varying amounts of L-methionine. Through the Ehrlich pathway, the rate and yield of 3-Met synthesis were constrained at low doses of L-methionine. However, at high L-methionine concentrations, yeast generated more 3-Met and accumulated more 3-methylthiopropionic acid, which was hazardous to the yeast itself [5]. The yeast changed regular metabolic pathways within the yeast cells by desulfurizing L-methionine to lower its effective concentration and prevent self-harm and excessive formation of 3-methylthiopropionic acid [3,13,45]. This ultimately resulted in a reduction in the capacity to synthesize 3-Met. The findings indicated that 4 g/L of L-methionine was the ideal concentration for *S. fibuligera* Y1402 to synthesize 3-Met. These findings were in line with those obtained for *H. burtonii* YHM-G and the majority of strains that have been published [3,7,13,46].

The influence of L-methionine on the synthesis of 3-Met by yeast was not only reflected in its concentration effect but also in the timing of its addition [7]. In contrast to *H. burtonii* YHM-G, the concentration of 3-Met synthesized by *S. fibuligera* Y1402 steadily declined with the delayed addition of L-methionine, as seen in Appendix A [7]. There are two possible causes for this: (1) L-methionine was added to the medium early in fermentation, which helped the yeast grow; (2) *S. fibuligera* Y1402 had a short lag phase, good environmental adaptability, and the ability to grow quickly in the medium under the conditions present. It is possible that this amino acid is necessary for the growth of *S. fibuligera* Y1402. Early in the fermentation process, *S. fibuligera* Y1402 may swiftly adapt to the Ehrlich pathway, supplying the energy or electrochemical gradient required for the production of 3-Met [7,13]. For these reasons, introducing L-methionine during the early stages of fermentation not only encouraged the development of *S. fibuligera* Y1402, but also gave the Ehrlich pathway enough time and circumstances to convert L-methionine into 3-Met. The differing needs for yeast extract concentration between *S. fibuligera* Y1402 and *H. burtonii* YHM-G may provide an explanation for this conjecture [7]. The concentration of yeast extract in the current medium was adequate to fulfill *S. fibuligera* Y1402’s demands for the synthesis of 3-Met, allowing for the organism’s fast growth and provision of the energy or electrochemical gradient required for the synthesis of 3-Met. But insufficient yeast extract in the first fermentation medium prevented *H. burtonii* YHM-G from growing, and this organism lacks the energy and electrochemical gradient needed to convert L-methionine into 3-Met. Therefore, this yeast may be able to synthesize 3-Met if L-methionine is added later [7]. This difference was mainly due to the inherent differences in the characteristics of the two yeasts.

Temperature is a crucial parameter that significantly affects microbial growth. When the temperature is too low, it decreases the fluidity of microbial cell membranes and the enzymatic activity, resulting in slow growth or even inhibition of microbial growth; conversely, when the temperature is too high, it can lead to the destruction of microbial cell structure and the inactivation of enzymes, causing microbial death [47]. Therefore, only suitable temperatures are favorable for the growth and metabolism of microorganisms, allowing them to accumulate more target products. *S. fibuligera* Y1402 was capable of producing 3-Met over a wide range of temperatures, with the most favorable temperature range for its synthesis being between 28 and 36 °C, as shown in Figure 3c. This was consistent with the optimal growth temperature range of *S. fibuligera* Y1402 and with the reported optimal temperature range for yeast synthesis of 3-Met in most studies [1,7].

pH is another important parameter that regulates microbial growth. It affects microbial growth and metabolic activities by altering the ion state of nutrients in the medium, the membrane potential of microbial cell membranes, the membrane permeability, and the functional activity of membrane proteins [26,40]. Through analysis, it was found that the initial pH had a significant impact on the yield of 3-Met by *S. fibuligera* Y1402 (Figure 3d). At an initial pH of 3-3.5, the yield of 3-Met was low, below 2.5 g/L. As the initial pH increased, the concentration of 3-Met produced by *S. fibuligera* Y1402 steadily increased. At an initial pH of 4.5-5, *S. fibuligera* Y1402 produced the highest amount of 3-Met, exceeding 3.2 g/L. Subsequently, the yield of 3-Met decreased with increasing pH, and at pH above 6.5, the yield of 3-Met was extremely low. This was similar to *H. burtonii* YHM-G, which also originated from the Baijiu fermentation environment [7]. It was possible that both yeasts had undergone long-term adaptation to the slightly acidic environment of Baijiu fermentation, resulting in the optimal pH of the enzymes involved in the Ehrlich pathway in these two yeasts being slightly acidic [1,7,13]. This can be inferred from the results of pH and biomass after fermentation under different conditions.

The inoculum size affects the rate of microbial growth and duration of fermentation cycles and has an important impact on the accumulation of target products and production costs [48]. Although the highest content of 3-Met produced by *S. fibuligera* Y1402 was observed at an inoculum size of 0.8%, overall, it was capable of a high yield of 3-Met within the range of a 0.1% to 3.2% inoculum size, as shown in Appendix A. This indicated that *S. fibuligera* Y1402 had good environmental adaptability or that the initial fermentation medium’s nutrient composition and content were suitable for the growth of *S. fibuligera* Y1402. However, when the inoculum size reached 6.4%, the content of 3-Met produced by *S. fibuligera* Y1402 decreased. This might be due to the yeast growing and reproducing too quickly, consuming more nutrients, and causing a severe lack of oxygen in the fermentation system. As a result, the strain’s ability to synthesize 3-Met was compromised. This was similar to the fermentation of 3-Met by *H. burtonii* YHM-G, which also required a similar inoculum size [7].

Based on previous research, it was known that yeast metabolizes L-methionine into 3-Met through the Ehrlich pathway, which is an aerobic process. Under conditions where the oxygen content is suitable, it may promote the expression of enzymes involved in the Ehrlich pathway, leading to the production of higher concentrations of 3-Met, even up to 10 times higher than under conditions of insufficient oxygen [1,7,13]. In the shake flask fermentation, the shaking speed and liquid volume were two important parameters that affected the oxygen content. The 3-Met concentration synthesized by *S. fibuligera* Y1402 at a speed of 45 rpm, as shown in Figure 3e, exhibited no significant variation when compared with static fermentation. However, this concentration was much greater under other speed parameters than under static fermentation. Additionally, the synthesis of 3-Met increased with increasing shaking speed, reaching a maximum of 3.9 g/L at a shaking speed of 225 rpm, which was approximately 10 times higher than under static conditions. This indicated that the yeast benefited from higher oxygen content for the synthesis of 3-Met. This was further confirmed by optimizing the liquid volume condition, as the highest concentration of 3-Met was achieved at lower liquid volumes. However, as the speed continued to increase, the yield of 3-Met by *S. fibuligera* Y1402 showed a slight decreasing trend. This was related to the dual effect of strong shear forces generated at high speeds causing mechanical damage to yeast cells and the higher production of 3-methylthiopropionic acid under high oxygen content causing chemical harm to yeast cell [6,7]. From the perspective of speed, the optimal speed for 3-Met production by *S. fibuligera* Y1402 was similar to that of *S. cerevisiae* S288C-CYS3 and higher than the optimal speed required by *H. burtonii* YHM-G [6,7]. From the perspective of liquid volume, the optimal liquid volume for *S. fibuligera* Y1402 was lower than that of *H. burtonii* YHM-G, indicating that *S. fibuligera* Y1402 required higher oxygen content for 3-Met production compared with *H. burtonii* YHM-G [7]. This might be due to the differences in their ability to resist physical shear forces and chemical toxicity, as well as the differences in the oxygen requirements of metabolism-related enzymes.

The optimal fermentation time is crucial for maximizing the profitability of industrial production using microorganisms, aiming to achieve the maximum accumulation of target products in the shortest time possible. The optimal fermentation time varies depending on the microbial species, target metabolites, fermentation medium, and cultivation conditions and requires targeted exploration. As shown in Figure 3f, *S. fibuligera* Y1402 was able to start converting L-methionine into 3-Met at the initial stage of fermentation (12 h). When the fermentation time reached 24 h, the concentration of 3-Met reached over 70% of the maximum value. Compared with *H. burtonii* YHM-G, this yeast had the characteristic of rapidly converting L-methionine into 3-Met, which might explain why *S. fibuligera* Y1402 started adding L-methionine earlier than *H. burtonii* YHM-G [7]. This can also be confirmed by the pH and cell density change trends of the two strains. However, compared with the optimal fermentation time of 48 h for *H. burtonii* YHM-G, *S. fibuligera* Y1402 took a slightly longer time, 60 h, to reach the highest concentration of 3-Met [7]. This time was much shorter than the fermentation time required by *S. cerevisiae* SC408, which could be attributed to the differences in microbial strains as well as variations in medium composition and fermentation conditions [11]. 

#### 3.4.2. Plackett–Burman Design

Based on the results of single-factor experiments, it was determined that the optimal surfactant for *S. fibuligera* Y1402 was Tween 80, with an addition amount of 2 g/L. The optimal timing for adding L-methionine was 0 h. The impact of several other factors on the production of 3-Met was also analyzed. Without considering the interaction effects, the optimal levels of these factors were determined. However, in reality, there were interactions between many factors, so further optimization of these factors was needed. To simplify the subsequent optimization process, the factors were reduced by using a Plackett–Burman design. The factors that significantly affect the production of 3-Met by *S. fibuligera* Y1402 were selected, and a sequence of the effects of these variables was established. These factors include glucose concentration, yeast extract concentration, L-methionine concentration, temperature, initial pH, inoculum size, shanking speed, liquid volume, and fermentation time. Table 1 displays the results. The table shows that the range of the yield of 3-Met generated by *S. fibuligera* Y1402 was 0.50 g/L to 3.83 g/L. The yeast yielded the largest amount of 3-Met in the tenth group when all factors—aside from yeast extract concentration—were at the ideal values found in the single-factor studies. This suggested that the interaction effects between the factors may not be significant. It is worth noting that although there was no significant difference between the maximum 3-Met production in the Plackett–Burman design experiment and the highest production in the single-factor experiment result, the two fermentation conditions for the highest 3-Met production were different. There was a lower concentration of yeast extract in the Plackett–Burman design experiment compared with in the single-factor experiment, which reduced the cost of 3-Met production.

Using the acquired findings, a multiple linear regression equation was fitted and encoded. The resulting equation is as follows:Y = 2.44 + 0.0291 A + 0.2849 B + 0.2421 C + 0.2191 D + 0.3153 E+ 0.2992 F+ 0.2528 G − 0.0574 H + 0.2224 I

The variance analysis of the results yielded a *p* value of 0.0134 (*p* < 0.05), indicating that the model was significant. The coefficient of determination (*R^2^*) for the regression equation was 0.9658, further demonstrating the reliability of the model and its ability to predict the results well. From the regression equation coefficients (Table 1), it was inferred that the liquid volume had a negative effect on the synthesis of 3-Met by yeast, while the other eight factors had positive effects. Through the analysis of the significance of each factor, it was evident that initial pH, fermentation time, yeast extract concentration, inoculum size, L-methionine concentration, shaking speed, and temperature had significant effects on the synthesis of 3-Met by yeast. The significance order of these factors followed the current writing order, indicating the need for further optimization research. On the other hand, the liquid volume and glucose concentration did not have a significant impact, so these two factors should be set at their optimal levels as determined by the single-factor experiment.

#### 3.4.3. Steepest Ascent Test

Combining the results of the single-factor experiments, Plackett–Burman design, practical experience, and principles of the steepest ascent test, the direction and step size of each factor in the steepest ascent test were determined. Results obtained based on this design are shown in Table 2. From these results, it was observed that the concentration of 3-Met synthesized by *S. fibuligera* Y1402 initially increased and then decreased with the increase in the experimental group. Under the conditions of the third experimental group, i.e., initial pH 5, fermentation time 52 h, yeast extract concentration 0.6 g/L, inoculum size 0.8%, L-methionine concentration 5 g/L, shaking speed 210 rpm, and temperature 32 °C, the highest yield of 3-Met synthesized by *S. fibuligera* Y1402 was 3.73 g/L. It was worth noting that this highest level was lower than the highest level obtained in the Plackett–Burman design, which might be due to the suboptimal step size design for each factor. However, as the difference between the two was not significant, it was decided to conduct subsequent response surface analysis using the factor levels corresponding to the third experimental group in the steepest ascent test.

#### 3.4.4. Response Surface Analysis

Based on the analysis of the Plackett–Burman design results, the significance of various factors affecting the production of 3-Met by *S. fibuligera* Y1402 was determined. Fermentation time, yeast extract concentration and initial pH were the three parameters that were chosen. The levels of the three factors were set as the third group in the steepest ascent test. According to the Box–Behnken experimental design principle, a total of 15 experiments with 3 factors and 3 levels were conducted, as shown in Table 3. The yield of 3-Met ranged from 1.66 to 4.09 g/L. The software Design-Expert 11.0 was used to perform a multivariate regression analysis and fit a multivariate quadratic equation regression to the above results. The regression coded equation obtained is as follows:Y = 3.95 + 0.5339A + 0.4763B+ 0.2444C − 0.8623A^2^ − 1.13B^2^ − 0.2833C^2^ − 0.9223AB + 0.0206 AC − 0.1504BC

From the analysis of variance, it was concluded that the pattern was *p* < 0.05, indicating that the model was significant. The *R^2^* of the quadratic regression equation was 0.9891, indicating that the equation fitted the experiment well with small errors. The insignificance of the lack-of-fit term implied a good correlation between the actual and predicted values. The yield of 3-Met was reflected and predicted by this model. The variation of the dependent variable depends on the changes in the independent variables, and the impact of the independent variables on the dependent variable can be reflected through the significance in the analysis of variance. In the regression equation, the *p* values of the first-order terms A (fermentation time), B (yeast extract concentration), and C (initial pH) were all less than 0.05, indicating that these three factors had a significant linear relationship with the synthesis of 3-Met by *S. fibuligera* Y1402 (Table 5). The order of influence of the three factors on the yield of 3-Met was inconsistent with the results of the Plackett–Burman tests, with B > C > A. This was related to differences in principles, i.e., the number and levels of factors involved in the two designs. The interaction terms AC and BC were significant, while AB was not significant. In the quadratic terms, the terms A^2^, B^2^, and C^2^ were all significant (*p* < 0.05), indicating significant curved effects between the fermentation time, yeast extract concentration, initial pH, and yield of 3-Met.

Based on the regression equation, a three-dimensional graph of the pairwise interaction effects between the three factors was plotted. Figure 4a indicates that the yield of 3-Met showed a clear increasing followed by a decreasing trend with increasing yeast extract concentration or fermentation time. The yield of 3-Met synthesized by *S. fibuligera* Y1402 increased initially and then decreased with increasing initial pH, but the overall trend was not significant, as shown in Figure 4b. However, with fermentation time, there was a significant increase followed by a decrease. According to Figure 4c, the yield of 3-Met also showed an initial increase followed by a decrease with increasing initial pH, but the impact was also not significant. However, with increasing yeast extract concentration, there was a clear increase followed by a decrease.

The regression equation was then used to determine the optimal conditions for maximizing the 3-Met yield by setting the partial derivatives of the equation to zero with respect to the independent variables. From the regression equation, it was concluded that the highest yield of 3-Met produced by *S. fibuligera* Y1402 occurred at an initial pH of 5.3, fermentation time of 54 h, and yeast extract concentration of 0.63 g/L, with a predicted value of 4.10 g/L. To validate this, three repeated experiments were conducted under the same conditions, and the yield of 3-Met produced by the yeast was found to be 4.03 g/L, which was very close to the predicted value with a deviation of only 1.7%, proving the reliability of the model. Compared with the reported studies (Table 6), the yield of 3-Met produced by *S. fibuligera* Y1402 was less only than that produced by the genetically modified strain *S. cerevisiae* S288C-AR010 and higher than that produced by all reported natural strains, making it the highest natural strain to produce 3-Met currently reported [18].

### 3.5. Scale-Up Fermentation in a 3 L Fermenter

Based on the above optimized conditions, the production of 3-Met by Y1402 in a 3 L fermenter was investigated. As the fermentation duration was extended, as seen in Figure 5, the concentration of 3-Met generated by Y1402 rose and peaked at 32–40 h. It produced the maximum yield of 3-Met at this point in the logarithmic growth phase, which was earlier than the phase for this strain and *H. burtonii* YHM-G in the shaking flask. This might be because of the strain’s characteristics and the varied oxygen supply [7]. The difference in the two yeasts’ results supported the earlier theory about optimizing the moment at which L-methionine is added. The yield of 3-Met decreased as fermentation continued. This could be because the cell used the 3-Met as a source of nutrition and transformed it into other substances because there were not enough nutrients available, or it could be because the cell died and prevented the 3-Met from being synthesized further, causing the 3-Met that had already been produced to be lost in the exhaust gas. Unfortunately, the yield of 3-Met was not ideal through scale-up culture, mainly due to the large amount of 3-Met lost in the exhaust gas, similar to the ethyl acetate produced reported in our previous study (Appendix A). In the future, other types of fermenters, such as centrifugal fermenters, will be used to explore the scaled-up cultivation with the hope of improving the 3-Met yield of Y1402.

### 3.6. Production of Aromatic Compounds

*S. fibuligera* Y1402 produced 28 kinds of flavor substances in SHM, including 8 alcohols, 3 esters, 2 aldehydes, 6 phenols, 2 acids, 3 furanones, 1 azole, and 3 other flavor substances (Table 7). A comparative analysis with the original flavor substances in the SHM revealed that only four flavor substances, phenethyl alcohol, (E)-9-octadecenoic acid ethyl ester, dimethyl phthalate, and 2,4-dimethyl-benzaldehyde, were present before and after fermentation. A total of 5 flavor substances were found only in the SHM, while 24 flavor substances were present in the post-fermentation medium. This indicated that *S. fibuligera* Y1402 converted the original flavor substances in the medium using specific metabolic pathways as well as produced new flavor substances. In addition, among the four common flavor substances, phenethyl alcohol and 2,4-dimethyl-benzaldehyde in the post-fermentation medium were derived not only from the original components of medium but also from the utilization of nutrients by *S. fibuligera* Y1402 [35]. Through analysis, it was found that the flavor substances produced by this yeast were mainly composed of compounds with floral, sweet, creamy, roasted nuts, and clove-like aromas. Representative flavor substances included (Z)-3,7-dimethyl-3,6-octadien-1-ol,2-methyl-1-propanol, (R)-2-octanol, 3-methyl-1-butanol, phenylethyl alcohol, 4-ethyl-2-methoxy-phenol, 2-methoxy-4-vinylphenol, and 2,3-dihydro-benzofuran. These flavor substances were in line with the current trend of a light and sweet style in Baijiu development, as well as the roasted aroma characteristic of sesame-flavored Baijiu, which may be related to the yeast originating from the environment of sesame-flavored Baijiu brewing [50,51]. It was worth noting that, consistent with the aforementioned optimization results, *S. fibuligera* Y1402 cannot synthesize 3-Met through a de novo synthesis pathway when there was no or very little L-methionine available. This once again confirmed that *S. fibuligera* Y1402 produced 3-Met through the Ehrlich pathway [7]. Therefore, in fermentation products that required the synthesis of 3-Met by *S. fibuligera* Y1402, a certain amount of L-methionine needed to be present in the fermentation product matrix or achieved by constructing a workable synthetic microbial community [19].

## 4. Conclusions

*S. fibuligera*, a high-yield 3-Met yeast, was selected from high-temperature Daqu. The *S. fibuligera* Y1402 strain showed an extensive growth pH range; a high tolerance to sugars, NaCl, nicotine, and 3-Met; and the ability to grow at temperatures as high as 40 °C. The optimal conditions for 3-Met synthesis by *S. fibuligera* Y1402 were determined by a single-factor experiment, Plackett–Burman design, steepest ascent test, and response surface analysis, which included a glucose concentration of 40 g/L, yeast extract concentration of 0.63 g/L, Tween 80 concentration of 2 g/L, initial pH of 5.3, fermentation time of 54 h, liquid volume of 25 mL/250 mL, inoculum size of 0.8%, L-methionine concentration of 5 g/L, shaking speed of 210 rpm, and temperature of 32 °C. Under these conditions, the yield of 3-Met reached 4.03 g/L, which is the highest among reported natural strains. In addition, the flavor substances produced by *S. fibuligera* Y1402 were mainly composed of compounds with floral, sweet, creamy, roasted nuts, and clove-like aromas, which have good application prospects in Baijiu brewing. In the future, researchers may focus on the construction of appropriate microbial flora with yeast Y1402 to increase the concentration of 3-Met in fermented foods like Baijiu, or production superior Daqu using the strain to boost the standard to Baijiu production. 

## Figures and Tables

**Figure 1 foods-13-00418-f001:**
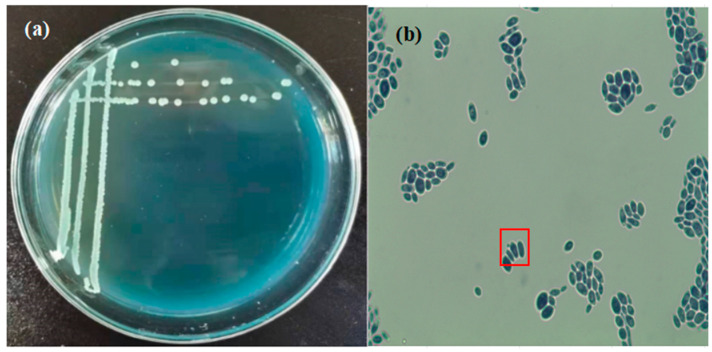
Colony morphological characteristics of yeast Y1402 on the WL identification medium (**a**) and cell morphological characteristics under microscopy (10 × 100) (**b**). The asexual budding reproduction occurred at the ends of the cells was highlighted in the red square.

**Figure 2 foods-13-00418-f002:**
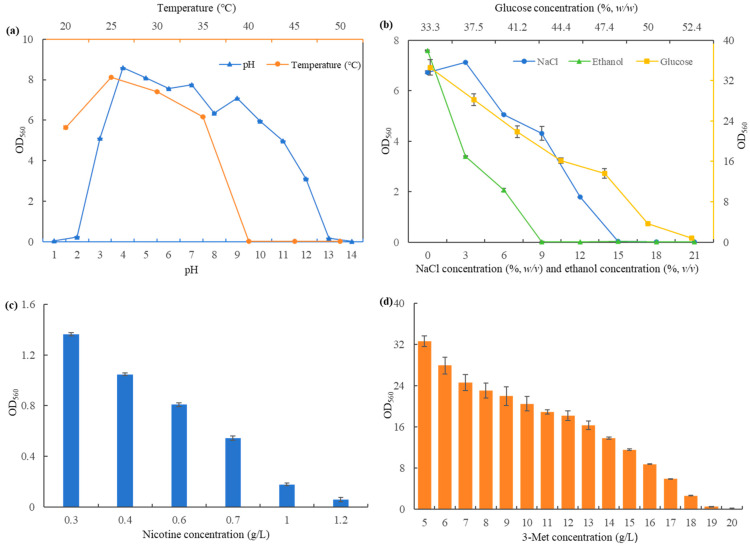
The range of growth temperature and pH (**a**); tolerance of glucose, NaCl, and ethanol (**b**); nicotine (**c**); and 3-Met (**d**) of strain Y1402.

**Figure 3 foods-13-00418-f003:**
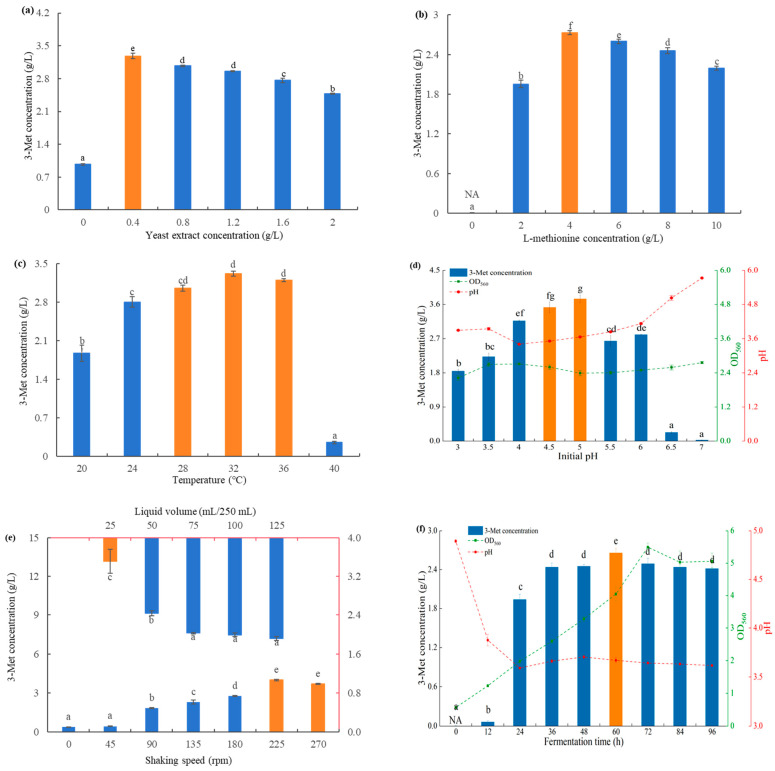
Effect of yeast extract concentration (0, 0.4, 0.8, 1.2, 1.6, and 2.0 g/L) (**a**), L-methionine concentration (0, 2, 4, 6, 8, and 10 g/L) (**b**), temperature (20, 24, 28, 32, 36, and 40 ℃) (**c**), initial pH (3.0, 3.5, 4.0, 4.5, 5.0, 5.5, 6.0, 6.5, and 7.0) (**d**), shaking speed ((0, 45, 90, 135, 180, 225 and 270 rpm) and liquid volume (25, 50, 75, 100, and 125 mL/250 mL) (**e**), and fermentation time (0, 12, 24, 36, 48, 60, 72, 84, and 96 h) (**f**) on 3-Met concentration. The same letters in the column indicate that the data do not differ significantly at 5% probability using Tukey’s test. The orange bar means the best conditions on 3-Met concentration. Note: the letter NA represents no detection of 3-Met.

**Figure 4 foods-13-00418-f004:**
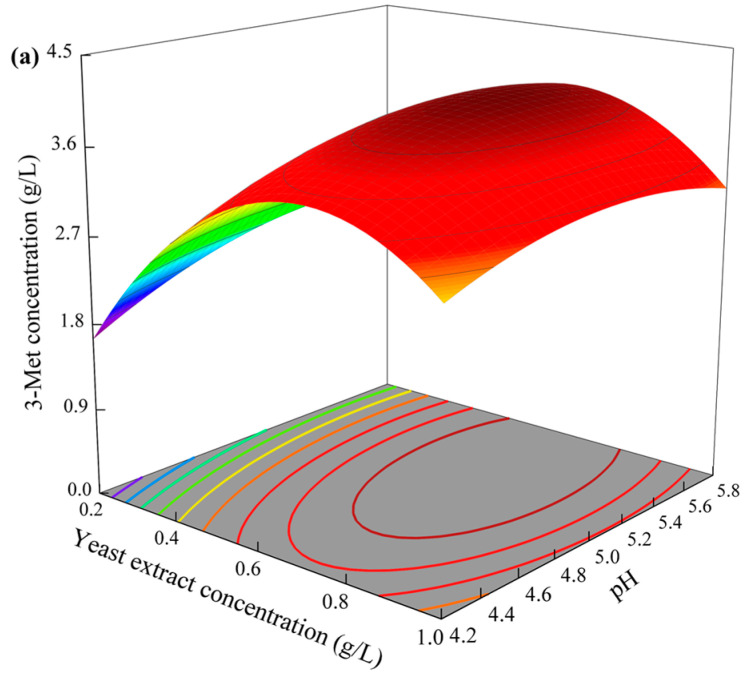
The three-dimensional graph of variables (fermentation time, yeast extract concentration, and initial pH) on the 3-Met concentration response using the Box–Behnken design. (**a**) Fermentation time and yeast extract concentration; (**b**) fermentation time and initial pH; (**c**) yeast extract concentration and initial pH.

**Figure 5 foods-13-00418-f005:**
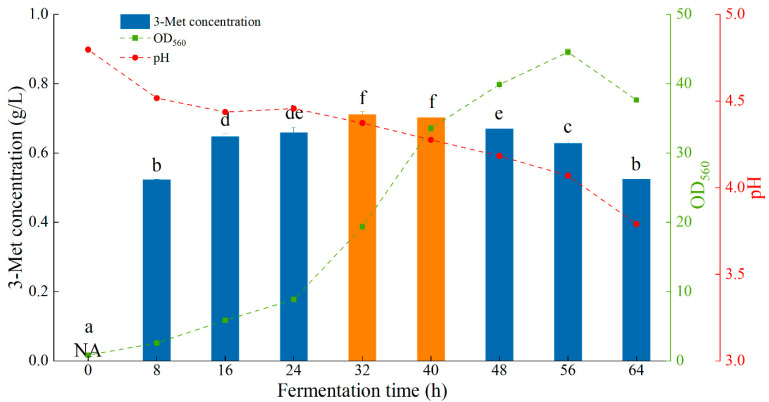
The result of 3-Met production by yeast Y1402 in a 3 L fermenter. The same letters in the column indicate that the data do not differ significantly at 5% probability using Tukey’s test. The orange bar means the best conditions on 3-Met concentration. Note: the letter NA represents no detection of 3-Met.

**Table 1 foods-13-00418-t001:** Plackett–Burman design for evaluating factors influencing 3-Met production and the statistical analysis.

NO.	Factors									Y: 3-Met Concentration (g/L)
	A: Glucose Concentration (g/L)	B: Yeast Extract Concentration (g/L)	C: L-Methionine Concentration (g/L)	D: Temperature (°C)	E: Initial pH	F: Fermentation Time (h)	G: Inoculum Size (%)	H: Liquid Volume (mL/250 mL)	I: Shaking Speed (rpm)	
1	50 (+1)	0.4 (−1)	6 (+1)	36 (+1)	4.5 (−1)	60 (+1)	0.4 (−1)	12.5 (−1)	180 (−1)	2.34
2	30 (−1)	0.4	2 (−1)	28 (−1)	4.5	48 (−1)	0.4	12.5	180	0.50
3	40 (0)	0.6 (0)	4 (0)	32 (0)	5 (0)	54 (0)	0.8 (0)	25 (0)	225 (0)	3.65
4	30	0.4	6	36	5.5 (+1)	48	1.2 (+1)	37.5 (+1)	180	2.45
5	50	0.4	2	28	5.5	60	1.2	12.5	270 (+1)	2.74
6	30	0.4	2	36	5.5	60	0.4	37.5	270	2.75
7	40	0.6	4	32	5	54	0.8	25	225	3.81
8	50	0.4	6	28	4.5	48	1.2	37.5	270	2.13
9	50	0.4	2	36	4.5	48	0.4	37.5	270	1.90
10	40	0.4	4	32	5	54	0.8	25	225	3.83
11	50	0.8 (+1)	6	28	5.5	60	0.4	37.5	180	2.73
12	30	0.8	6	28	5.5	48	0.4	12.5	270	2.89
13	30	0.8	2	28	4.5	60	1.2	37.5	180	2.32
14	50	0.8	2	36	5.5	48	1.2	12.5	180	2.96
15	30	0.8	6	36	4.5	60	1.2	12.5 (−1)	270	3.54
Effect (Exi)	0.00291	0.712	0.121	0.0548	0.631	0.02493	0.632	−0.00459	0.00494	
*F* values	0.19	18.54	13.38	10.96	22.7	20.43	14.59	0.75	11.29	
*p* values	0.683	0.013	0.022	0.03	0.009	0.011	0.019	0.435	0.028	
Rank	9	3	5	7	1	2	4	8	6	
Significance	-	*	*	*	**	*	*	-	*	

Note: “-”, not significant (*p* > 0.05); “*”, significant at the 5% level (*p* < 0.05); “**”, significant at the 1% level (*p* < 0.01).

**Table 2 foods-13-00418-t002:** Experimental designs and results of the steepest ascent test for 3-Met production.

Groups	Initial pH	Fermentation Time (h)	Yeast Extract Concentration (g/L)	Inoculum Size (%)	L-Methionine Concentration (g/L)	Shaking Speed (rpm)	Temperature (°C)	3-Met Concentration (g/L)
1	4.5	36	0.2	0.4	3	180	28	2.28
2	4.75	44	0.4	0.6	4	195	30	3.13
3	5	52	0.6	0.8	5	210	32	3.73
4	5.25	60	0.8	1.0	6	225	34	3.62
5	5.5	68	1.0	1.2	7	240	36	2.66

**Table 3 foods-13-00418-t003:** Box–Behnken design and responses of the dependent variables.

NO.	Fermentation Time (h)		Yeast Extract Concentration (g/L)		Initial pH		3-Met Concentration (g/L)
	A	Code A	B	Code B	C	Code C	Y
1	60	1	1.0	1	5.8	1	1.83
2	44	−1	1.0	1	4.2	−1	2.51
3	60	1	0.6	0	4.2	−1	3.15
4	44	−1	0.6	0	5	0	2.61
5	60	1	1.0	1	5	0	2.09
6	52	0	0.2	−1	4.2	−1	1.66
7	44	−1	0.6	0	5.8	1	2.41
8	52	0	0.6	0	5	0	3.83
9	52	0	0.2	−1	5.8	1	2.53
10	60	1	0.2	−1	5	0	2.89
11	52	0	1.0	1	5.8	1	3.19
12	52	0	0.6	0	5	0	3.91
13	52	0	0.6	0	5	0	4.09
14	52	0	1.0	1	4.2	−1	2.83
15	44	−1	1.0	1	5	1	2.83

Note: the letter Y represents the 3-Met concentration.

**Table 4 foods-13-00418-t004:** Strains for 3-Met production in the present study.

Strain Number	3-Met Production (mg/L)	Strain Number	3-Met Production (mg/L)	Strain Number	3-Met Production (mg/L)	Strain Number	3-Met Production (mg/L)
B1	0 ± 0	B25	0 ± 0	B49	24.57 ± 1.03	Y12	217.73 ± 3.26
B2	0 ± 0	B26	0 ± 0	B50	0 ± 0	Y13	0 ± 0
B3	0 ± 0	B27	0 ± 0	B51	0 ± 0	Y14	895.29 ± 5.09
B4	2.94 ± 0.22	B28	0 ± 0	B52	0 ± 0	Y1402	2001 ± 10.37
B5	0 ± 0	B29	45.78 ± 2.12	B53	0 ± 0	Y15	0 ± 0
B6	0 ± 0	B30	0 ± 0	B54	0 ± 0	Y16	0 ± 0
B7	53.85 ± 4.01	B31	0 ± 0	B55	0 ± 0	Y17	1131 ± 9.27
B8	5.52 ± 1.38	B32	0 ± 0	B56	0 ± 0	M1	39.88 ± 2.03
B9	30.57 ± 4.91	B33	6.33 ± 0.24	B57	34.65 ± 6.36	M2	0 ± 0
B10	0 ± 0	B34	0 ± 0	B58	0 ± 0	M3	0 ± 0
B11	0 ± 0	B35	0 ± 0	B59	0 ± 0	M4	0 ± 0
B12	0 ± 0	B36	0 ± 0	B60	52.49 ± 7.18	M5	0 ± 0
B13	0 ± 0	B37	0 ± 0	B61	39.09 ± 0.95	M6	0 ± 0
B14	0 ± 0	B38	42.61 ± 2.00	Y1	0 ± 0	M7	0 ± 0
B15	0 ± 0	B39	0 ± 0	Y2	1411.63 ± 17.39	M8	0 ± 0
B16	0 ± 0	B40	0 ± 0	Y3	1464.34 ± 8.46	M9	0 ± 0
B17	0 ± 0	B41	41.20 ± 1.73	Y4	1021.86 ± 5.19	M10	43.63 ± 2.28
B18	0 ± 0	B42	0 ± 0	Y5	1208.04 ± 17.30	M11	54.58 ± 3.03
B19	0 ± 0	B43	0 ± 0	Y6	1690.45 ± 14.22	M12	30.39 ± 8.26
B20	0 ± 0	B44	0 ± 0	Y7	0 ± 0	M13	0 ± 0
B21	52.73 ± 6.02	B45	0 ± 0	Y8	115.09 ± 2.97	M14	34.83 ± 6.17
B22	14.93 ± 3.17	B46	0.73 ± 0.16	Y9	559.45 ± 7.13	M15	52.69 ± 2.04
B23	49.32 ± 0.74	B47	49.16 ± 3.36	Y10	1488.61 ± 12.05		
B24	0 ± 0	B48	0 ± 0	Y11	0 ± 0		

Note: The letter B represents bacteria; Y represents yeast; M represents mold.

**Table 5 foods-13-00418-t005:** Regression coefficients and their significances for 3-Met production from the results of the Box–Behnken design.

Source	Sum of Squares	DF	Mean Square	*F* Values	*p* Values	Significance
Model	7.22573	9	0.80286	49.06	0.000	***
A—Fermentation time	1.23168	1	1.23168	75.26	0.000	***
B—Yeast extract concentration	1.27371	1	1.27371	77.83	0.000	***
C—Initial pH	0.37036	1	0.37036	22.63	0.005	**
A^2^	0.22626	1	0.22626	13.83	0.014	*
B^2^	2.41522	1	2.41522	147.58	0.000	***
C^2^	2.21996	1	2.21996	135.65	0.000	***
AB	0.00101	1	0.00101	0.06	0.814	-
AC	0.10892	1	0.10892	6.66	0.049	*
BC	1.92710	1	1.92710	117.75	0.000	***
Residual	0.08183	5	0.01637			
Lack of Fit	0.04553	3	0.01518	0.84	0.585	-
Pure Error	0.03630	2	0.01815			
Cor Total	7.30756	14				
	*R*^2^ = 0.9891	*R*^2^_Adj_ = 0.9694				

Note: “-”, not significant (*p* > 0.05); “*”, significant at the 5% level (*p* < 0.05); “**”, significant at the 1% level (*p* < 0.01); “***”, significant at the 0.1% level (*p* < 0.001).

**Table 6 foods-13-00418-t006:** Summary of 3-Met production by strains with a content exceeding 0.5 g/L.

Strain Classify	Strain	Culture Institution	L-Met Concentration (g/L)	3-Met Production (mg/L)	Reference
Natural strain	*S. cerevisiae* S288C	Beijing Technology and Business University, Beijing 100048, China	10	590	[14]
	*S. cerevisiae* SC408	Beijing Technology and Business University, Beijing 100048, China	4	1600	[11]
	*H. burtonii* YHM-G	Beijing Technology and Business University, Beijing 100048, China	6	3160	[7]
	*K. lactis* KL71	National University of Singapore, Singapore 117543	-	990	[13]
	*S. cerevisiae* SC57	Beijing Technology and Business University (BTBU), Beijing 100048, China	4	1600 (fed-batch fermentation without D101); 2260 (fed-batch fermentation with D101)	[12]
	*S. fibuligera* Y1402	Beijing Technology and Business University (BTBU), Beijing 100048, China	5	4030	This study
Engineered strain	*S. cerevisiae* S288C-CYS3	Beijing Technology and Business University, Beijing 100048, China	4	690	[6]
	*S. cerevisiae* s288c-ARO10	Beijing Technology and Business University, Beijing 100048, China	10	900	[16]
	*S. cerevisiae* C3	Beijing Technology and Business University, Beijing 100048, China	10	600	[49]
	S. cerevisiae S288C	Beijing Technology and Business University, Beijing 100048, China	10	940	[49]
	*S. cerevisiae* AR8	Beijing Technology and Business University, Beijing 100048, China	10	760	[14]
	*S. cerevisiae* S288C-AR010	Beijing Technology and Business University, Beijing 100048, China	5	4380	[18]
	*S. cerevisiae* CEN.PK113-7D	Biochemical Engineering Group, Theodor-Heuss-Allee 25, 60486 Frankfurt am Main, Germany		2200	[5]
	*S. cerevisiae* (co-expression of ARO8 and ARO100	School of Food and Chemical Engineering, Beijing Technology and Business University, Beijing 100048, China	10	3240	[17]

**Table 7 foods-13-00418-t007:** The volatile compounds in SHM with or without Y1402 (µg/L).

Volatile Compounds	SHM ^a^	Y1402 ^b^	Aroma Descriptors
(Z)-3,7-Dimethyl-3,6-octadien-1-ol	- ^c^	280 ± 35	Floral
2-Methyl-1-propanol	-	33 ± 5	Sweet, wine
3-Methyl-3-buten-1-ol	-	5 ± 1	Slightly apple-like
(R)-2-Octanol	-	300 ± 23	Creamy, cucumber
3-Methyl-1-butanol	-	444 ± 51	Floral
1-Octen-3-ol	-	4 ± 2	Mushroom
2-Propyl-1-pentanol	-	3 ± 1	Mushroom, cream
Phenylethyl alcohol	19 ± 3	304 ± 57	Rosy, honey
∑Alcohols	19	1373	
Dibutyl phthalate	-	30 ± 11	Odorless
(E)-9-Octadecenoic acid ethyl ester	3 ± 1	5 ± 1	Lipid
Dimethyl phthalate	4 ± 1	2 ± 0	Odorless
Hexadecanoic acid ethyl ester	6 ± 0	-	Cream, herb
Ethyl linoleate	2 ± 0	-	Lipid
∑Esters	15	37	
2,4-Dimethyl-benzaldehyde	14 ± 2	85 ± 13	Semen armeniacae amarae
Furfural	-	3 ± 0	Roasted, almond
∑Aldehydes	14	88	
4-Ethyl-phenol	-	15 ± 6	Horsy, barnyard, smoky, medical aromatic
2-Methoxy-phenol	-	13 ± 3	Castoreum, smoke, bacon, ham
4-Ethyl-2-methoxy-phenol	-	110 ± 33	Spicy, smoky, bacon, clove
Phenol	-	6 ± 2	Rubber
2-Methoxy-4-vinylphenol	-	920 ± 49	Fruity, clove
2,4-Di-tert-butylphenol	-	10 ± 1	Phenolic
∑Phenols	-	1074	
3-Methyl-butanoic acid	-	8 ± 2	Sour, cheesy
2-Methyl-propanoic acid	-	5 ± 1	Cheese, dairy, buttery
∑Acids	-	13	
2,3-Dihydro-benzofuran	-	356 ± 52	Roast, sweet
Dihydro-5-pentyl-2(3H)-furanone	-	11 ± 2	Coconut, creamy, sweet, buttery, oily
5-Hexyldihydro-2(3H)-furanone	-	11 ± 4	Fruity, creamy, peach, coconut, buttery, sweet
∑Furanone	-	378	
1,2-Benzisothiazole	-	2 ± 0	N ^d^
Benzothiazole	83 ± 6	-	Meaty, vegetative
∑Azoles	83	2	
7,9-Di-tert-butyl-1-oxaspiro(4,5)deca-6,9-diene-2,8-dione	-	6 ± 0	N
Pentadecane	-	2 ± 0	Waxy
(3Z,5E)-1,3,5-Undecatriene	-	7 ± 3	Oily
Aniline	5 ± 0	-	Sweet
Methyl-pyrazine	2 ± 0	-	Roast
∑Others	7	15	
Sum	138	2980	

Note: “^a^”, the volatile compounds in SHM without Y1402; “^b^”, the volatile compounds in SHM with Y1402; “^c^”, not detected; “^d^”, no information.

## Data Availability

Data are contained within the article or Appendix A.

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
