# Peer review of "Screening, Identification, and Fermentation Condition Optimization of a High-Yield 3-Methylthiopropanol Yeast and Its Aroma-Producing Characteristics"

_foods, 2024, doi:10.3390/foods13030418_

Round 1
Reviewer 1 Report
Comments and Suggestions for Authors
This study is on " Identification and fermentation condition optimization of a high-yield 3-methylthiopropanol yeast". The study explored medium composition and fermentation conditions via 3-stage sequential experiments. One factor at a time provided the set of factors with the domain. Plackett-Burman design identified seven key factors for 3-Met production. Box-Behnken design determined optimal conditions, achieving a 4.03 g/L 3-Met yield. RSM was used in this case. The methodology seems sound and the topic is of great interest. However, there are certain concerns as mentioned below:
L38-53: It can be reduced in line with the scope of the work.
L80-81: Please discuss relevant literature, show the research gap, and establish the novelty.
L85: Please spell out the objective clearly.
L101-116: Is it an established method or modified? You can just mention the reference and refrain from elaborating on it.
L158-186: It can be shown in the form of a schematic. How were the lower and upper limits for each factor decided? Did the author validate the OD at 600 nm with an actual microbial population count? What is the relation between them?
L204-207: How did the results of single factor analysis lead to the selection of the Plackett-Burman (PB) design?
L219: Is it the right word to use “gradient” here?
L224: Based on which parameter, 3 variables were selected?
L235-237: How the optimization was conducted?
L414: The caption is not clear. Please rewrite.
Sec 3.4.1: Single factor optimization: From a statistical point of view, this has no meaning. Because, in every step of the experiment, the baseline has been changed.
L562: Please don’t say it as optimal. It is rather a selection from the single-factor analysis. How did single-factor experiments help to decide PB design? How were the lower and upper limits for each factor decided in PB design?
L562-570: Please be crisp and to the point. It is not clear based on which parameter, PB variables were selected. If there is any screening after single factor analysis?
L577-583: From single factor analysis, you can’t ‘optimize’ the condition. You can only ‘select’ the best-performing condition. Optimization means it may be at conditions other than points where experiments have been conducted.
L586: Is it in coded form or actual form?
Table 5: How EXi was calculated?
Please check the p-values and significance in Table 5.
Table 1 and Table 5 can be clubbed while providing the key information from Table 5 into Table 1.
Sec 3.4.3: PB design confirmed no effect from ‘glucose concentration’ and “liquid volume”. In the steepest accent, these two were removed. PB design also confirmed the highest influence of Initial pH, fermentation time, and yeast conc. So, this should be your next step for BBD. It is not clear how the steepest accent helped in this matter.
What is the formula for expression or steepest accent used?
L629-630: Why only the second-order polynomial model has been fitted? You can do this in software but based on which parameter you discarded other models?
L632: Is it in coded form or real form?? In the manuscript, I do not see any equation of the conversion between real and coded forms of variables.
L648-649: Not clear. PB developed only a linear model.
In Table 6, only the coefficient and p-values are sufficient besides the model ANOVA data. You are not discussing other values from this table.
L666-667: How the optimization was performed? Please elaborate in detail with the expression and selection of relative importance values.
Table 7: Include the result from this study also.
Table 8: Is it possible to compare the data with any benchmark sample in terms of its final application?
L752-753: Please mention about PB and BBD suitably.
Author Response
20240116
Dear Reviewer,
Re: Manuscript ID. Foods-2805565 “Screening, identification, fermentation condition optimization of a high-yield 3-methylthiopropanol yeast and its aroma-producing characteristics”
Thank you very much for your interest in our manuscript, and for providing us with helpful suggestions for improving its quality. In the following, we have made modifications and replies point-by-point according to your suggestions. Revisions in the manuscript are highlighted in blue.
We look forward to hearing from you at your early convenience.
Yours sincerely,
Guangsen Fan
Address: School of Food and Health, Beijing Technology and Business University, Beijing 100048, China
Tel: +86 13811497684
E-mail: [email protected]
Responses to the comments
- This study is on " Identification and fermentation condition optimization of a high-yield 3-methylthiopropanol yeast". The study explored medium composition and fermentation conditions via 3-stage sequential experiments. One factor at a time provided the set of factors with the domain. Plackett-Burman design identified seven key factors for 3-Met production. Box-Behnken design determined optimal conditions, achieving a 4.03 g/L 3-Met yield. RSM was used in this case. The methodology seems sound and the topic is of great interest. However, there are certain concerns as mentioned below:
Response: Thank you very much for your recognition of our manuscript and for providing us with many valuable suggestions. We have made detailed revisions based on your comments, which are crucial for improving the quality of our manuscript.
- L38-53: It can be reduced in line with the scope of the work.
Response: Thank you for your advice, we have appropriately reduced the content of this section as per your suggestion.
Page 1-2, line 40-51:
3-Methylthiopropanol (3-Met) is a highly important sulfur-containing flavor com-pound that exhibits a low aroma threshold, presenting a pleasant meaty, grilled cheese, or sweet aroma at concentrations between 1-3 mg/L [1-3]. It is a widely used edible flavor compound, and its market demand is growing rapidly [1]. Owing to its cheap cost, chemical synthesis is now the primary method for producing 3-Met; however, this synthesis has disadvantages as well, such as the use of hazardous raw materials, pollution during synthesis process, and challenges in completely removing harmful byproducts [4]. These problems cannot meet the increasing consumer demand for environmentally friendly, green, healthy, and natural flavorings [4,5]. In this context, research efforts have shifted to the biotransformation synthesis, an ecologically safe, effective, and green approach, to produce 3-Met [6]. Future predictions indicate that it will take precedence over other methods for the commercial synthesis of natural 3-Met.
- L80-81: Please discuss relevant literature, show the research gap, and establish the novelty.
Response: Thank you very much for your comments. This part mainly aims to prove our stated viewpoint through relevant literature. By regulating the microorganisms that produce high levels of 3-Met in traditional fermented foods, we can increase the content of 3-Met in fermented foods, emphasizing the importance of screening for high-yielding 3-Met microorganisms from these brewing environments. It is not about introducing the novelty of our research by analyzing the differences between the literatures. Based on your comments, we have made appropriate modifications.
Page 2, line 79-81:
Consequently, it is essential to identify and get more natural strains from various fer-mented food brewing conditions that provide elevated level of 3-Met.
- L85: Please spell out the objective clearly.
Response: Thank you very much for your comments. We have revised our target to Baijiu.
Page 2, line 84-86:
This study aims to increase the 3-Met content in Baijiu, hence improving its overall quality. In addition, this study will offer valuable strains and theoretical frameworks applicable to related fields.
- L101-116: Is it an established method or modified? You can just mention the reference and refrain from elaborating on it.
Response: Thank you very much for your comments. The method is already established, so we have made the necessary modifications based on your suggestions, including adding references. To ensure the reproducibility of the experiments, we have retained some relevant information and eliminated irrelevant details.
Page 3, line 101-108:
The isolation of strains was performed according to Fan et al. using the traditional plate coating method with LB, YPD, and PDA medium [23]. All isolated strains were individually cultured in their respective liquid culture media at 30°C and 200 rpm. Bacteria and yeast were cultured for 36 hours, while molds were cultured for 60 hours. The cultured strains were adjusted to a cell or spore count of 1×106 per milliliter using a hemocytometer. The process of producing 3-Met included transferring 0.1 mL of the corrected culture to a 250 mL shake flask with 50 mL of the original fermentation me-dium and incubating it for 48 hours at 30°C at 200 rpm, as described by Ma et al [7].
- L158-186: It can be shown in the form of a schematic. How were the lower and upper limits for each factor decided? Did the author validate the OD at 600 nm with an actual microbial population count? What is the relation between them?
Response: Thank you very much for your comments. We can actually determine the upper and lower limits of growth through visual observation of growth or non-growth. However, for the purpose of measurement, we used turbidity method to determine the growth. Since we only want to evaluate the growth status under different conditions rather than determine the cell count under different conditions, we did not perform cell counting. Therefore, there is no relationship between turbidity and cell counting. Considering that there is no inherent connection between these parts and they are independent of each other, we have made appropriate deletions based on your comments to highlight the important information instead of a schematic.
Page 4, line 152-160:
After being activated, S. fibuligera Y1402 was introduced into YPD medium with a 0.2% inoculum size to assess its growth temperature (ranging from 20°C to 50°C), pH levels (ranging from 1 to 14), sugar tolerance (33.3%, 37.5%, 41.2%, 44.4%, 47.4%, 50%, and 52.4%, w/w), NaCl tolerance (0%, 3%, 6%, 9%, 12%, 15%, 18%, and 21%, w/v), ethanol tolerance (0%, 3%, 6%, 9%, 12%, 15%, 18%, and 21%, v/v), nicotine tolerance (0.3 g/L, 0.4 g/L, 0.6 g/L, 0.7 g/L, 1.0 g/L, and 1.2 g/L) and 3-Met tolerance (ranging from 5% to 20%). The cultures were incubated in a shaking incubator at 28°C and 180 rpm for 48 hours. The OD560 was measured using a turbidity method. Blank control was performed using uninoculated medium under various cultivation conditions.
- L204-207: How did the results of single factor analysis lead to the selection of the Plackett-Burman (PB) design?
Response: Thank you very much for your feedback. Our previous description was not comprehensive and accurate enough, so we have made the necessary modifications. We determined the factors and center point based on the significance analysis of single-factor experiments and experimental experience.
Page 5, line 179-183:
Nine factors (glucose concentration (A), yeast extract concentration (B), L-methionine concentration (C), temperature (D), initial pH (E), fermentation time (F), inoculum size (G), liquid volume (H), and shaking speed (I)) were chosen for optimization using the Plackett-Burman experimental design based on the significant difference analysis of the results of the single-factor experiment and experience.
- L219: Is it the right word to use “gradient” here?
Response: Thank you very much for your feedback. You are absolutely right, and we have made the necessary changes based on your comments.
Page 5, line 194-196:
The gradient for each factor was determined based on the principles of the steepest ascent test, single-factor experiment results, and experimental experience.
- L224: Based on which parameter, 3 variables were selected?
Response: We have selected the top three most important factors based on their impact on 3-Met in PB experiments. This is due to our previous misrepresentation, and we have made the necessary modifications.
Page 6, line 199-202:
Three components (initial pH, fermentation time, and yeast extract concentration) having the greatest influence were chosen for optimization utilizing the response sur-face analysis based on the significance order of factors impacting the yield of 3-Met from the Plackett-Burman design.
- L235-237: How the optimization was conducted?
Response: We did not conduct optimization studies, but rather drew from the experiences of other studies and our laboratory's previous related research. In fact, we initially used ventilation rate of 0.5vvm and rotational speed of 200 rpm, but the results were not satisfactory. Later, we referred to other studies and adjusted the conditions to 0.5vvm and rotational speed of 300 rpm, which produced slightly better results, but the increase in content was not significant.
- L414: The caption is not clear. Please rewrite.
Response: We are very sorry. We have made the necessary changes based on your suggestions.
Page 12, line 395-397:
The yeast extract concentration required to achieve optimum 3-Met synthesis by S. fibuligera Y1402 was comparable to that of K. lactis KL71 but lower than that of H. burtonii YHM-G, S. cerevisiae EC-1118, and S. cerevisiae SC408 [7,11,13,40].
- Sec 3.4.1: Single factor optimization: From a statistical point of view, this has no meaning. Because, in every step of the experiment, the baseline has been changed.
Response: Thank you very much. We used single factor optimization under fixed basic fermentation conditions to select the factors that significantly affect yeast production of 3-Met and determine the optimal level range of the factors, which can help with the design of PB experiments.
- L562: Please don’t say it as optimal. It is rather a selection from the single-factor analysis. How did single-factor experiments help to decide PB design? How were the lower and upper limits for each factor decided in PB design?
Response: Thank you very much for your feedback. Although under the premise of fixed basic fermentation conditions, it is not possible to obtain the optimal fermentation conditions, some experimental factors and their levels can be determined through single factor experiments. Through single factor experiments, we found that Tween 80 has a positive effect on the production of 3-Met by yeast Y1402. However, when analyzing different concentrations of Tween 80, we found that there was no significant difference in the production of 3-Met at different concentrations. Based on cost considerations, we determined the concentration of Tween 80. We also found that the timing of L-methionine addition can affect the production of 3-Met by yeast. As mentioned earlier, single factor experiments can determine the impact of each factor on the production of 3-Met by yeast, and can determine the optimal level conditions. Based on the significant differences in the impact of different levels of each factor on 3-Met and the optimal level conditions, we selected the upper and lower level values for each factor based on the center point of the PB experiment design.
- L562-570: Please be crisp and to the point. It is not clear based on which parameter, PB variables were selected. If there is any screening after single factor analysis?
Response: Thank you very much for your comments. As mentioned above, some experimental factors were determined based on actual experimental results, while most factors need to be determined based on the results of single factor experiments to design the PB experiment combinations according to the PB experimental design principles. The center point and upper and lower limit levels of PB experiments are determined based on the significant differences in the impact of different levels of each factor on the production of 3-Met. We hope that this explanation is helpful and accurate.
- L577-583: From single factor analysis, you can’t ‘optimize’ the condition. You can only ‘select’ the best-performing condition. Optimization means it may be at conditions other than points where experiments have been conducted.
Response: Thank you very much for your comments. As you mentioned, we did indeed proceed based on this principle, determining the factors and their level ranges for PB experiments through single-factor experiments. During this process, we were able to identify the levels of some experimental factors, such as the type of surfactant in our experimental conditions. In other researchers' studies, such as the types of carbon source, nitrogen source, and material particle size also can be determined. By analyzing the differences in single-factor experimental results, we could also identify factors that do not significantly affect 3-Met production at different levels, such as the concentration of Tween 80 in our experimental conditions. These analysis results will help us design PB experiments to more accurately identify the optimal fermentation conditions.
- L586: Is it in coded form or actual form?
Response: Thank you for your comments. We appreciate your feedback on our work. As you mentioned, we have previously used the "actual form" and have now modified it to the "coded form". We have also described this change in the manuscript. We hope that this clarification is helpful and accurate.
Page 15, line 567-570:
Using the acquired findings, a multiple linear regression equation was fitted and encoded. The resulting equation is as follows:
Y= 2.44 + 0.0291 A + 0.2849 B + 0.2421 C + 0.2191 D + 0.3153 E+ 0.2992 F+ 0.2528 G - 0.0574 H + 0.2224 I
- Table 5: How EXi was calculated?
Response: We appreciate your comments. In fact, it is the coefficients of different factors in the simulation equation.
- Please check the p-values and significance in Table 5.
Response: We appreciate your comments. Thank you for your verification. We have checked the data and confirmed that there are no issues. If you have any further questions or concerns, please let us know.
- Table 1 and Table 5 can be clubbed while providing the key information from Table 5 into Table 1.
Response: Thank you very much for your valuable feedback. We have made the changes as suggested, and indeed, the presentation is much improved compared to the previous version. Your comments have greatly helped us enhance the quality of our manuscript.
Page 5, line 186:
Table 1. Plackett-Burman Design for evaluating factors influencing 3-Met production and the statistical analysis.
NO. |
Factors |
Y:3-Met concentration (g/L) |
||||||||||||||||
A:Glucose concentration (g/L) |
B:Yeast extract concentration (g/L) |
C:L-methionine concentration (g/L) |
D: Temperature (°C) |
E:Initial pH |
F:Fermentation time (h) |
G:Inoculum size (%) |
H:Liquid volume (mL/250 mL) |
I:Shaking speed (rpm) |
||||||||||
1 |
50 (+1) |
0.4 (-1) |
6 (+1) |
36 (+1) |
4.5 (-1) |
60 (+1) |
0.4 (-1) |
12.5 (-1) |
180 (-1) |
2.34 |
||||||||
2 |
30 (-1) |
0.4 |
2 (-1) |
28 (-1) |
4.5 |
48 (-1) |
0.4 |
12.5 |
180 |
0.50 |
||||||||
3 |
40 (0) |
0.6 (0) |
4 (0) |
32 (0) |
5 (0) |
54 (0) |
0.8 (0) |
25 (0) |
225 (0) |
3.65 |
||||||||
4 |
30 |
0.4 |
6 |
36 |
5.5 (+1) |
48 |
1.2 (+1) |
37.5 (+1) |
180 |
2.45 |
||||||||
5 |
50 |
0.4 |
2 |
28 |
5.5 |
60 |
1.2 |
12.5 |
270 (+1) |
2.74 |
||||||||
6 |
30 |
0.4 |
2 |
36 |
5.5 |
60 |
0.4 |
37.5 |
270 |
2.75 |
||||||||
7 |
40 |
0.6 |
4 |
32 |
5 |
54 |
0.8 |
25 |
225 |
3.81 |
||||||||
8 |
50 |
0.4 |
6 |
28 |
4.5 |
48 |
1.2 |
37.5 |
270 |
2.13 |
||||||||
9 |
50 |
0.4 |
2 |
36 |
4.5 |
48 |
0.4 |
37.5 |
270 |
1.90 |
||||||||
10 |
40 |
0.4 |
4 |
32 |
5 |
54 |
0.8 |
25 |
225 |
3.83 |
||||||||
11 |
50 |
0.8 (+1) |
6 |
28 |
5.5 |
60 |
0.4 |
37.5 |
180 |
2.73 |
||||||||
12 |
30 |
0.8 |
6 |
28 |
5.5 |
48 |
0.4 |
12.5 |
270 |
2.89 |
||||||||
13 |
30 |
0.8 |
2 |
28 |
4.5 |
60 |
1.2 |
37.5 |
180 |
2.32 |
||||||||
14 |
50 |
0.8 |
2 |
36 |
5.5 |
48 |
1.2 |
12.5 |
180 |
2.96 |
||||||||
15 |
30 |
0.8 |
6 |
36 |
4.5 |
60 |
1.2 |
12.5 (-1) |
270 |
3.54 |
||||||||
Effect (Exi) |
0.00291 |
0.712 |
0.121 |
0.0548 |
0.631 |
0.02493 |
0.632 |
-0.00459 |
0.00494 |
|
||||||||
F values |
0.19 |
18.54 |
13.38 |
10.96 |
22.7 |
20.43 |
14.59 |
0.75 |
11.29 |
|
||||||||
P values |
0.683 |
0.013 |
0.022 |
0.03 |
0.009 |
0.011 |
0.019 |
0.435 |
0.028 |
|
||||||||
Rank |
9 |
3 |
5 |
7 |
1 |
2 |
4 |
8 |
6 |
|
||||||||
Significance |
- |
* |
* |
* |
** |
* |
* |
- |
* |
|
||||||||
Note: “-”, not significant (P > 0.05); “*”, significant at 5% level (P < 0.05); “**”, significant at 1% level (P < 0.01).
- Sec 3.4.3: PB design confirmed no effect from ‘glucose concentration’ and “liquid volume”. In the steepest accent, these two were removed. PB design also confirmed the highest influence of Initial pH, fermentation time, and yeast conc. So, this should be your next step for BBD. It is not clear how the steepest accent helped in this matter.
Response: Thank you for your comments. You raise an important point about the potential challenges with BBD when the central values are not appropriately chosen, leading to difficulties in obtaining the optimal level combination. The use of steepest accent to approach the optimal level region is indeed effective approach to identify the peak value as the center point for the response surface, enabling a more accurate optimization within the desired region and ultimately obtaining the optimal level combination. This approach ensures that the experiments are conducted in the most promising area to obtain the desired results.
- What is the formula for expression or steepest accent used?
Response: We appreciate your comments. You are correct in pointing out that the steepest ascent design involves conducting a limited number of experimental combinations with factors varying in a specific direction and gradient based on the results from PB experiments. By selecting the highest value from these limited experimental groups, one can effectively approach the optimal level region. This approach helps to identify the optimal level combination with greater precision and efficiency. Thank you for your feedback, it is very helpful in improving the quality of our work.
- L629-630: Why only the second-order polynomial model has been fitted? You can do this in software but based on which parameter you discarded other models?
Response: We appreciate your comments. You bring up an important point about the consideration of interactions and surface effect in response surface design. It is common practice to use quadratic polynomials for modeling, as they provide a high degree of simulation and the lack of significant lack of fitness suggests good predictive power for experimental values. While it is possible to choose higher or lower polynomial orders, the choice of a second-order polynomial is based on convention and is the default mode in many software packages.
- L632: Is it in coded form or real form?? In the manuscript, I do not see any equation of the conversion between real and coded forms of variables.
Response: We appreciate your comments. As you mentioned, we previously used the "real form" and have now converted it to the "coded form" as per your suggestion. We have also provided clarification in the manuscript regarding this change. Thank you for your feedback and assistance in improving the quality of our work.
Page 1, line 23-26:
Page 16, line 608-610:
The regression coded equation obtained is as follows:
Y = 3.95 + 0.5339A + 0.4763B+ 0.2444C -0.8623A2 – 1.13B2 – 0.2833C2 - 0.9223AB + 0.0206 AC - 0.1504BC
- L648-649: Not clear. PB developed only a linear model.
Response: We appreciate your comments. It may have been a matter of our wording. Through PB experiments, we found that the optimal level combination was determined by individual factors, suggesting that there was no significant interaction or the interaction was relatively low. Based on your advice, we recognize that our previous statement was incorrect, and we have removed it accordingly. Thank you for your feedback and assistance in improving the quality of our work.
Page 16, line 624-625:
The interaction terms AC and BC were significant, while AB was not significant.
- In Table 6, only the coefficient and p-values are sufficient besides the model ANOVA data. You are not discussing other values from this table.
Response: We appreciate your comments. As you mentioned, Table 6 mainly presents the coefficient and p-values, which are essential in assessing the reliability of the model and the significance of individual factors' impact on the response value. However, the other parameters in the table serve as necessary elements for presenting these two key parameters. Therefore, we have retained them in the table. Thank you for your feedback.
- L666-667: How the optimization was performed? Please elaborate in detail with the expression and selection of relative importance values.
Response: Thank you very much for your comments. We did not state clearly. We have made modifications. It is mainly obtained by calculating the derivative of the simulation formula. Of course, we used the function modules provided by the software to calculate.
Page 17, line 641-643:
The regression equation was then used to determine the optimal conditions for maximizing the 3-Met yield by setting the partial derivatives of the equation to zero with respect to the independent variables.
- Table 7: Include the result from this study also.
Response: Thank you very much for your comments. We have made the following additions.
Page 18, line 662:
Table 6. Summary of 3-Met production by strains with a content exceeding 0.5 g/L.
Strain classify |
Strain |
Culture institution |
L-Met concentration (g/L) |
3-Met production (mg/L) |
Reference |
Natural strain |
S. cerevisiae S288C
|
Beijing Technology and Business University, Beijing 100048, China |
10 |
590 |
[14] |
S. cerevisiae SC408
|
Beijing Technology and Business University, Beijing 100048, China |
4 |
1600 |
[11] |
|
H. burtonii YHM-G
|
Beijing Technology and Business University, Beijing 100048, China |
6 |
3160 |
[7] |
|
K. lactis KL71 |
National University of Singapore, Singapore 117543 |
- |
990 |
[13] |
|
S. cerevisiae SC57 |
Beijing Technology and Business University (BTBU), Beijing 100048, China |
4 |
1600 (fed-batch fermentation without D101); 2260 (fed-batch fermentation with D101) |
[12] |
|
S. fibuligera Y1402 |
Beijing Technology and Business University (BTBU), Beijing 100048, China |
5 |
4030 |
This study |
|
Engineered strain |
S. cerevisiae S288C-CYS3 |
Beijing Technology and Business University, Beijing 100048, China |
4 |
690 |
[6] |
S. cerevisiae s288c-ARO10
|
Beijing Technology and Business University, Beijing 100048, China
|
10 |
900 |
[16] |
|
S. cerevisiae C3
|
Beijing Technology and Business University, Beijing 100048, China |
10 |
600 |
[49] |
|
S. cerevisiae S288C |
Beijing Technology and Business University, Beijing 100048, China |
10 |
940 |
[49] |
|
S. cerevisiae AR8 |
Beijing Technology and Business University, Beijing 100048, China |
10 |
760 |
[14] |
|
S. cerevisiae S288C-AR010
|
Beijing Technology and Business University, Beijing 100048, China |
5 |
4380 |
[18] |
|
S. cerevisiae CEN.PK113-7D |
Biochemical Engineering Group, Theodor-Heuss-Allee 25, 60486 Frankfurt am Main, Germany |
|
2200 |
[5] |
|
S. cerevisiae (co-expression of ARO8 and ARO100 |
School of Food and Chemical Engineering, Beijing Technology and Business University, Beijing 100048, China |
10 |
3240 |
[17] |
- Table 8: Is it possible to compare the data with any benchmark sample in terms of its final application?
Response: Thank you very much for your opinion. In fact, this is based on the application of Baijiu. We choose the fragrance producing medium as the basic medium commonly used in the research of Baijiu. Through the difference analysis of the volatile flavor substances between the inoculated strains and the non-inoculated strains, we found that this yeast can produce a variety of composed of compounds with floral, sweet, creamy, toasted nuts, and clone like aroma. It plays an important role in developing and improving the sweet flavor of Baijiu. In the later stage, we will conduct relevant research based on the application of the strain in Baijiu, such as the impact on the quality of Baijiu by strengthening Daqu or directly adding it to fermented grains.
- L752-753: Please mention about PB and BBD suitably.
Response: Thank you very much for your opinion. We have made the following additions.
Page 22, line 718-724:
The optimal conditions for 3-Met synthesis by S. fibuligera Y1402 were determined by a single-factor experiment, Plackett-Burman design, steepest ascent test and response surface analysis, which included a glucose concentration of 40 g/L, yeast extract con-centration of 0.63 g/L, Tween 80 concentration of 2 g/L, initial pH of 5.3, fermentation time of 54 h, liquid volume of 25 mL/250 mL, inoculum size of 0.8%, L-methionine concentration of 5 g/L, shaking speed of 210 rpm, and temperature of 32°C.
Reviewer 2 Report
Comments and Suggestions for Authors
By the present study, the authors utilized the conventional plate coating screening method to obtain a high-yield 3-methylthiopropanol (3-Met) yeast Y1402 from sesame-flavored Daqu based on the flavor characteristics of 3-Met and people's preferences for natural products. Morphological observation, physiological, biochemical parameters (temperature, pH etc), molecular biology methods were adopted to identify the yeast as Saccharomycopsis fibuligera. The composition medium was optimized, and the maximum response region determined. Finally, aroma analysis revealed that the flavor substances produced by S. fibuligera Y1402 were mainly composed of compounds with floral, sweet, creamy, roasted nut, and clove-like aromas. Therefore, based on this wide analysis, the authors conclude that S. fibuligera showed great potential for application in fermented foods.
In this reviewer’s opinion, this interesting study is well conducted as concerns the number of analysis carried out and consequently the amount of data reported, as well as for the analytical handling of measures and results. All these items were also enriched by an excellent graphical representation.
The minor remarks concern:
1) the Data Analysis (paragraph 2.6 ) that would be better named Statistical Analysis and the Statistical methods adopted that are reported in the text specifically reported, and briefly described and/or justified;
2) The novelty and even more the limitation of the study (never mentioned) would be better reported in a separate paragraph.
Comments on the Quality of English Language
Minor editing required
Author Response
20240116
Dear Reviewer,
Re: Manuscript ID. Foods-2805565 “Screening, identification, fermentation condition optimization of a high-yield 3-methylthiopropanol yeast and its aroma-producing characteristics”
Thank you very much for your interest in our manuscript, and for providing us with helpful suggestions for improving its quality. In the following, we have made modifications and replies point-by-point according to your suggestions. Revisions in the manuscript are highlighted in blue.
We look forward to hearing from you at your early convenience.
Yours sincerely,
Guangsen Fan
Address: School of Food and Health, Beijing Technology and Business University, Beijing 100048, China
Tel: +86 13811497684
E-mail: [email protected]
Responses to the comments
By the present study, the authors utilized the conventional plate coating screening method to obtain a high-yield 3-methylthiopropanol (3-Met) yeast Y1402 from sesame-flavored Daqu based on the flavor characteristics of 3-Met and people's preferences for natural products. Morphological observation, physiological, biochemical parameters (temperature, pH etc), molecular biology methods were adopted to identify the yeast as Saccharomycopsis fibuligera. The composition medium was optimized, and the maximum response region determined. Finally, aroma analysis revealed that the flavor substances produced by S. fibuligera Y1402 were mainly composed of compounds with floral, sweet, creamy, roasted nut, and clove-like aromas. Therefore, based on this wide analysis, the authors conclude that S. fibuligera showed great potential for application in fermented foods.
In this reviewer’s opinion, this interesting study is well conducted as concerns the number of analysis carried out and consequently the amount of data reported, as well as for the analytical handling of measures and results. All these items were also enriched by an excellent graphical representation.
Response: Thank you very much for your recognition of the relevant work in our manuscript. We deeply appreciate your comments and have responded to each of them.
The minor remarks concern:
- the Data Analysis (paragraph 2.6 ) that would be better named Statistical Analysis and the Statistical methods adopted that are reported in the text specifically reported, and briefly described and/or justified;
Response: Thank you very much for your comments. We have made modifications based on your suggestions.
Page 7, line 226-228:
2.7. Statistical Analysis
A one-way ANOVA (p < 0.05) with Tukey’s test was used to examine statistical differences in the evaluated techniques.
- The novelty and even more the limitation of the study (never mentioned) would be better reported in a separate paragraph.
Response: Thank you very much. The novelty of our work lies in the acquisition of a new strain of high-yielding 3-Met. This strain can also produce a variety of flavor compounds. Of course, the shortcoming is that it requires L-methionine as a precursor to produce high-yielding 3-Met, which is mentioned in our manuscript. Based on your comments, we have reviewed the conclusion section, highlighting the key points, while the shortcomings are not specifically highlighted in the conclusion, but indirectly reflected through the relevant work we will carry out in the future..
Page 20-21, line 707-710:
It was worth noting that, consistent with the aforementioned optimization results, S. fibuligera Y1402 cannot synthesize 3-Met through de novo synthesis pathway when there was no or very little L-methionine available.
Page 22, line 727-734
Under these conditions, the yield of 3-Met reached 4.03 g/L, which is the highest among reported natural strains. In addition, the flavor substances produced by S. fibuligera Y1402 were mainly composed of compounds with floral, sweet, creamy, roasted nuts, and clove-like aromas, which have good application prospects in Baijiu brewing. In the future, researchers may focus on the construction of appropriate microbial flora with yeast Y1402 to increase the concentration of 3-Met in fermented foods like Baijiu and using the strain to produce superior Daqu to boost the standard to Baijiu production.
Reviewer 3 Report
Comments and Suggestions for Authors
Subject: Peer Review of Manuscript ID [foods-2805565], titled "Identification and fermentation condition optimization of a high-yield3-methylthiopropanol yeast"
Dear Authors,
I have completed a thorough review of your manuscript submitted to Foods Journal. Your study on the optimization of fermentation conditions for high-yield 3-methylthiopropanol-producing yeast presents significant insights. However, there are several areas that require attention and improvement.
I have detailed the major concerns in the attached review document, which includes issues related to clarity, methodology, result interpretation, data presentation, and overall structure. Specific recommendations have been provided for each identified issue.
The study has the potential to contribute significantly to the field, but it is crucial to address these concerns to enhance the clarity, reproducibility, and impact of your research. I recommend a major revision of the manuscript.
Thank you for the opportunity to review your work, and I look forward to seeing the revised manuscript.
Sincerely,
*****
I have completed the review of the research paper from Foods Journal. Here are major issues identified:
- Title and Abstract Clarity: The title and abstract should more precisely reflect the specific focus of the study, including the emphasis on aromatic compound analysis.
- Literature Review Depth: The introduction requires a more comprehensive review of relevant literature, especially on the biochemical pathways of 3-methylthiopropanol production.
- Methodology Detail: The materials and methods section needs more detailed descriptions of the experimental procedures for reproducibility.
- Strain Selection Justification: A more thorough rationale for the selection of the particular yeast strain should be provided.
- Statistical Analysis: The statistical methods used in analyzing experimental data need clarification and justification.
- Results Interpretation: The discussion of results requires a deeper analysis, especially in the context of existing research.
- Control Experiments: The paper should include more information about control experiments to validate the results.
- Data Presentation: Graphs and tables need to be more clearly presented and interpreted.
- Experimental Replicability: Ensure the experiments described are replicable with the provided information.
- Citation Consistency: Ensure all citations are consistent and correctly formatted.
- Language and Grammar: The paper requires thorough proofreading to correct grammatical and typographical errors.
- Conclusion Specificity: The conclusion section should more specifically summarize the study's findings and implications.
- Future Work: Suggestions for future research directions are somewhat vague and could be more specific.
Based on these issues, I recommend major revision before the paper is suitable for publication.
Comments on the Quality of English LanguageAdditionally, the manuscript would benefit significantly from a thorough review for language and grammar. Proper editing for English language usage will enhance the clarity and readability of the paper, ensuring that the scientific content is effectively communicated to the journal's international audience.
Author Response
20240116
Dear Reviewer,
Re: Manuscript ID. Foods-2805565 “Screening, identification, fermentation condition optimization of a high-yield 3-methylthiopropanol yeast and its aroma-producing characteristics”
Thank you very much for your interest in our manuscript, and for providing us with helpful suggestions for improving its quality. In the following, we have made modifications and replies point-by-point according to your suggestions. Revisions in the manuscript are highlighted in blue.
We look forward to hearing from you at your early convenience.
Yours sincerely,
Guangsen Fan
Address: School of Food and Health, Beijing Technology and Business University, Beijing 100048, China
Tel: +86 13811497684
E-mail: [email protected]
- Subject: Peer Review of Manuscript ID [foods-2805565], titled "Identification and fermentation condition optimization of a high-yield3-methylthiopropanol yeast"
Dear Authors,
I have completed a thorough review of your manuscript submitted to Foods Journal. Your study on the optimization of fermentation conditions for high-yield 3-methylthiopropanol-producing yeast presents significant insights. However, there are several areas that require attention and improvement.
I have detailed the major concerns in the attached review document, which includes issues related to clarity, methodology, result interpretation, data presentation, and overall structure. Specific recommendations have been provided for each identified issue.
The study has the potential to contribute significantly to the field, but it is crucial to address these concerns to enhance the clarity, reproducibility, and impact of your research. I recommend a major revision of the manuscript.
Thank you for the opportunity to review your work, and I look forward to seeing the revised manuscript.
Sincerely,
*****
I have completed the review of the research paper from Foods Journal. Here are major issues identified:
Response: Thank you very much for your recognition of our research, and thank you for the important comments you have made, which are of great significance in improving the quality of our manuscript. We have carefully revised and responded to your comments.
- Title and Abstract Clarity:The title and abstract should more precisely reflect the specific focus of the study, including the emphasis on aromatic compound analysis.
Response: Thank you very much for your comments. We have revised the title and abstract based on your suggestions.
Page 1, line 2-4:
Screening, identification, fermentation condition optimization of a high-yield 3-methylthiopropanol yeast and its aroma-producing characteristics
Page 1, line 13-26:
A high-yield 3-methylthiopropanol (3-Met) yeast Y1402 was obtained from sesame-flavored Daqu and it was identified as Saccharomycopsis fibuligera. S. fibuligera Y1402 showed a broad range of growth temperatures and pH, and the maximum tolerance to glucose, NaCl, nicotine, and 3-Met at 50% (w/w), 15% (w/v), 1.2 g/L, and 18 g/L, respectively. After optimization by single-factor experiments, Plackett-Burman design, steepest ascent test and Box-Behnken design, the 3-Met yield reached 4.03 g/L by S. fibuligera Y1402 under the optimal conditions that glucose concentration was 40 g/L, yeast extract concentration was 0.63 g/L, Tween 80 concentration was 2 g/L, L-methionine concentration was 5 g/L, liquid volume was 25 mL/250 mL, initial pH was 5.3, fermentation temperature was 32°C, inoculum size was 0.8%, shaking speed was 210 rpm, and fermentation time was 54 h. The fermentation was scaled up to a 3 L fermenter under the optimized conditions, and the yield of 3-Met reached 0.71 g/L. Additionally, aroma analysis revealed that the flavor substances produced by S. fibuligera Y1402 in sorghum hydrolysate medium mainly composed of compounds with floral, sweet, creamy, roasted nut, and clove-like aromas. There-fore, S. fibuligera had great potential for application in the brewing of Baijiu and other fermented foods.
- Literature Review Depth:The introduction requires a more comprehensive review of relevant literature, especially on the biochemical pathways of 3-methylthiopropanol production.
Response: Thank you very much for your comments. Given that the main content of our manuscript is the screening, identification, and optimization of conditions for high-yielding 3-Met microorganisms, rather than the study of its formation mechanism, we have appropriately supplemented the key genes for its biochemical synthesis.
Page 2, line 52-54:
To maximize the output of 3-Met synthesis, these naturally existing strains were modified by overexpressing or shutting down genes, such as aminotransferase genes ARO8 and ARO9 and decarboxylase gene ARO10 [14-17].
- Methodology Detail:The materials and methods section needs more detailed descriptions of the experimental procedures for reproducibility.
Response: Thank you very much for your comments. Based on your comments and those of other experts, we have made modifications to the method section.
Page 5, line 170-174:
Nine factors (glucose concentration (A), yeast extract concentration (B), L-methionine concentration (C), temperature (D), initial pH (E), fermentation time (F), inoculum size (G), liquid volume (H), and shaking speed (I)) were chosen for optimi-zation using the Plackett-Burman experimental design based on the significant differ-ence analysis of the results of the single-factor experiment and experience.
Page 6, line 190-193:
Three components (initial pH, fermentation time, and yeast extract concentration) having the greatest influence were chosen for optimization utilizing the response sur-face analysis based on the significance order of factors impacting the yield of 3-Met from the Plackett-Burman design.
Page 6, line 218-222:
A one-way ANOVA (p < 0.05) with Tukey’s test was used to examine statistical differences in the evaluated techniques. Each experiment was performed in triplicate, and the experimental data were processed and plotted using Excel 2019 (Microsoft, Redmond, WA, USA), SPSS 24.0 (IBM Corp., New York, NY, USA), OriginPro 9.1 (OriginLab, Northampton, MA, USA) and Design-Expert 11 (Stat-Ease, Inc., Minneap-olis, MN, USA).
- Strain Selection Justification:A more thorough rationale for the selection of the particular yeast strain should be provided.
Response: Thank you very much for your comments. Our main goal in this study is to obtain a strain with higher production of 3-Met. Therefore, we selected the strain with the highest yield in the initial screening as the indicator, and did not consider other factors.
- Statistical Analysis:The statistical methods used in analyzing experimental data need clarification and justification.
Response: Thank you very much for your comments. We have added relevant analysis methods.
Page 6, line 218-222:
A one-way ANOVA (p < 0.05) with Tukey’s test was used to examine statistical differences in the evaluated techniques. Each experiment was performed in triplicate, and the experimental data were processed and plotted using Excel 2019 (Microsoft, Redmond, WA, USA), SPSS 24.0 (IBM Corp., New York, NY, USA), OriginPro 9.1 (OriginLab, Northampton, MA, USA) and Design-Expert 11 (Stat-Ease, Inc., Minneap-olis, MN, USA).
- Results Interpretation:The discussion of results requires a deeper analysis, especially in the context of existing research.
Response: Thank you very much for your comments. We have added and clarified some of the discussions.
Page 12, line 399-401:
The effect of surfactants on the formation of 3-Met by H. burtonii YHM-G, on the other hand, was different and might have resulted from small differences in the yeast cell’s membrane composition or the activity of associated Ehrlich pathway enzymes [8].
Page 12, line 411-413:
Through the Ehrlich pathway, 3-Met might be produced from L-methionine [8,44]. In summary, it undergoes transamination, decarboxylation, and reduction processes, 3-Met was produced from L-methionine [5].
- Control Experiments:The paper should include more information about control experiments to validate the results.
Response: Thank you very much for your comments. We have added more information about control experiments.
Page 4, line 150-151:
Blank control was performed using uninoculated medium under various cultivation conditions.
Page 4, line 159-160:
A control group devoid of yeast cells was used. The previously disclosed method was used to treat the SHM medium.
- Data Presentation:Graphs and tables need to be more clearly presented and interpreted.
Response: Thank you very much for your comments. We have made appropriate adjustments to some of the charts and tables in order to present the relevant content clearly.
Page 5, line 177:
Table 1. Plackett-Burman Design for evaluating factors influencing 3-Met production and the statistical analysis.
NO. |
Factors |
Y:3-Met concentration (g/L) |
||||||||||||||||
A:Glucose concentration (g/L) |
B:Yeast extract concentration (g/L) |
C:L-methionine concentration (g/L) |
D: Temperature (°C) |
E:Initial pH |
F:Fermentation time (h) |
G:Inoculum size (%) |
H:Liquid volume (mL/250 mL) |
I:Shaking speed (rpm) |
||||||||||
1 |
50 (+1) |
0.4 (-1) |
6 (+1) |
36 (+1) |
4.5 (-1) |
60 (+1) |
0.4 (-1) |
12.5 (-1) |
180 (-1) |
2.34 |
||||||||
2 |
30 (-1) |
0.4 |
2 (-1) |
28 (-1) |
4.5 |
48 (-1) |
0.4 |
12.5 |
180 |
0.50 |
||||||||
3 |
40 (0) |
0.6 (0) |
4 (0) |
32 (0) |
5 (0) |
54 (0) |
0.8 (0) |
25 (0) |
225 (0) |
3.65 |
||||||||
4 |
30 |
0.4 |
6 |
36 |
5.5 (+1) |
48 |
1.2 (+1) |
37.5 (+1) |
180 |
2.45 |
||||||||
5 |
50 |
0.4 |
2 |
28 |
5.5 |
60 |
1.2 |
12.5 |
270 (+1) |
2.74 |
||||||||
6 |
30 |
0.4 |
2 |
36 |
5.5 |
60 |
0.4 |
37.5 |
270 |
2.75 |
||||||||
7 |
40 |
0.6 |
4 |
32 |
5 |
54 |
0.8 |
25 |
225 |
3.81 |
||||||||
8 |
50 |
0.4 |
6 |
28 |
4.5 |
48 |
1.2 |
37.5 |
270 |
2.13 |
||||||||
9 |
50 |
0.4 |
2 |
36 |
4.5 |
48 |
0.4 |
37.5 |
270 |
1.90 |
||||||||
10 |
40 |
0.4 |
4 |
32 |
5 |
54 |
0.8 |
25 |
225 |
3.83 |
||||||||
11 |
50 |
0.8 (+1) |
6 |
28 |
5.5 |
60 |
0.4 |
37.5 |
180 |
2.73 |
||||||||
12 |
30 |
0.8 |
6 |
28 |
5.5 |
48 |
0.4 |
12.5 |
270 |
2.89 |
||||||||
13 |
30 |
0.8 |
2 |
28 |
4.5 |
60 |
1.2 |
37.5 |
180 |
2.32 |
||||||||
14 |
50 |
0.8 |
2 |
36 |
5.5 |
48 |
1.2 |
12.5 |
180 |
2.96 |
||||||||
15 |
30 |
0.8 |
6 |
36 |
4.5 |
60 |
1.2 |
12.5 (-1) |
270 |
3.54 |
||||||||
Effect (Exi) |
0.00291 |
0.712 |
0.121 |
0.0548 |
0.631 |
0.02493 |
0.632 |
-0.00459 |
0.00494 |
|
||||||||
F values |
0.19 |
18.54 |
13.38 |
10.96 |
22.7 |
20.43 |
14.59 |
0.75 |
11.29 |
|
||||||||
P values |
0.683 |
0.013 |
0.022 |
0.03 |
0.009 |
0.011 |
0.019 |
0.435 |
0.028 |
|
||||||||
Rank |
9 |
3 |
5 |
7 |
1 |
2 |
4 |
8 |
6 |
|
||||||||
Significance |
- |
* |
* |
* |
** |
* |
* |
- |
* |
|
||||||||
Note: “-”, not significant (P > 0.05); “*”, significant at 5% level (P < 0.05); “**”, significant at 1% level (P < 0.01).
Page 11, line 360:
Figure 3. Effect of yeast extract concentration (0, 0.4, 0.8, 1.2, 1.6, and 2.0 g/L) (a), L-methionine concentration (0, 2, 4, 6, 8, and 10 g/L) (b), temperature (20, 24, 28, 32, 36 and 40 ℃) (c), initial pH (3.0, 3.5, 4.0, 4.5, 5.0, 5.5, 6.0, 6.5 and 7.0) (d), shaking speed ((0, 45, 90, 135, 180, 225 and 270 rpm) and liquid volume (25, 50, 75, 100 and 125 mL/250 mL) (e), and fermentation time (0, 12, 24, 36, 48, 60, 72, 84 and 96 h) (f) on 3-Met concentration. Same letters in the column indicates that the data do not differ significantly at 5% probability by the Tukey test.
- Experimental Replicability:Ensure the experiments described are replicable with the provided information.
Response: Thank you very much for your comments. Our experiments are all three-fold parallel experiments, and we have conducted multiple repeated verification tests on key factors and levels. And this research has been transformed into applications in tobacco flavor enhancement, and has achieved good results, ensuring the reproducibility of the experiment.
Page 7, line 219-222:
Each experiment was performed in triplicate, and the experimental data were processed and plotted using Excel 2019 (Microsoft, Redmond, WA, USA), SPSS 24.0 (IBM Corp., New York, NY, USA), OriginPro 9.1 (OriginLab, Northampton, MA, USA) and Design-Expert 11 (Stat-Ease, Inc., Minneapolis, MN, USA).
- Citation Consistency:Ensure all citations are consistent and correctly formatted.
Response: We are very sorry for the problem with the format of the references. We have made modifications to ensure consistency and accuracy in the format.
- Language and Grammar:The paper requires thorough proofreading to correct grammatical and typographical errors.
Response: We are very sorry, and we have corrected and revised the language of the manuscript from beginning to end according to your suggestions.
- Conclusion Specificity:The conclusion section should more specifically summarize the study's findings and implications.
Response: Thank you very much for your comments. We have made appropriate modifications to this.
Page 22, line 709-724:
- fibuligera, a high-yield 3-Met yeast, was selected from high-temperature Daqu. The S. fibuligera Y1402 strain showed an extensive growth pH range, a high tolerance to sugars, NaCl, nicotine, and 3-Met, and the ability to grow at temperatures as high as 40°C. The optimal conditions for 3-Met synthesis by S. fibuligera Y1402 were determined by a single-factor experiment, Plackett-Burman design, steepest ascent test and response surface analysis, which included a glucose concentration of 40 g/L, yeast extract concentration of 0.63 g/L, Tween 80 concentration of 2 g/L, initial pH of 5.3, fermentation time of 54 h, liquid volume of 25 mL/250 mL, inoculum size of 0.8%, L-methionine concentration of 5 g/L, shaking speed of 210 rpm, and temperature of 32°C. Under these conditions, the yield of 3-Met reached 4.03 g/L, which is the highest among reported natural strains. In addition, the flavor substances produced by S. fibuligera Y1402 were mainly composed of compounds with floral, sweet, creamy, roasted nuts, and clove-like aromas, which have good application prospects in Baijiu brewing. In the future, researchers may focus on the construction of appropriate microbial flora with yeast Y1402 to increase the concentration of 3-Met in fermented foods like Baijiu and using the strain to produce superior Daqu to boost the standard to Baijiu production.
- Future Work:Suggestions for future research directions are somewhat vague and could be more specific.
Response: Thank you very much for your comments. We have supplemented and specified the future research on the production of 3-Met around this strain.
Page 22, line 721-72:
In the future, researchers may focus on the construction of appropriate microbial flora with yeast Y1402 to increase the concentration of 3-Met in fermented foods like Baijiu and using the strain to produce superior Daqu to boost the standard to Baijiu production.
- Based on these issues, I recommend major revision before the paper is suitable for publication.
Response: Thank you very much for your valuable comments, and thank you for giving us the opportunity to revise and improve the quality of the manuscript.
- Comments on the Quality of English Language
Additionally, the manuscript would benefit significantly from a thorough review for language and grammar. Proper editing for English language usage will enhance the clarity and readability of the paper, ensuring that the scientific content is effectively communicated to the journal's international audience.
Response: Thank you very much for your comments. We have revised and improved the language of the manuscript.

Round 2
Reviewer 1 Report
Comments and Suggestions for Authors
The authors have answered all the queries raised by the reviewers satisfactorily.
Author Response
The authors have answered all the queries raised by the reviewers satisfactorily.
Response: Dear Reviewer, thank you very much for your help. The improvement of our manuscript quality and its acceptance are attributed to your valuable guidance. Thank you again.
Reviewer 3 Report
Comments and Suggestions for Authors
Dear Authors,
The paper can be now published in its current form.
Congratulations!
Author Response
Dear Authors,
The paper can be now published in its current form.
Congratulations!
Response: Dear Reviewer, thank you for your congratulations. We are deeply grateful for your guidance and assistance.